# Concomitant opening of a bulk-gap with an emerging possible Majorana zero mode

Anna Grivnin[1,2], Ella Bor[1,2], Moty Heiblum[1,2], Yuval Oreg[2] & Hadas Shtrikman[1,2]

Majorana quasiparticles are generally detected in a 1D topological superconductor by tunneling electrons into its edge, with an emergent zero-bias conductance peak (ZBCP). However, such a ZBCP can also result from other mechanisms, hence, additional verifications are required. Since the emergence of a Majorana must be accompanied by an opening of a topological gap in the bulk, two simultaneous measurements are performed: one in the bulk and another at the edge of a 1D InAs nanowire coated with epitaxial aluminum. Only under certain experimental parameters, a closing of the superconducting bulk-gap that is followed by its reopening, appears simultaneously with a ZBCP at the edge. Such events suggest the occurrence of a topologically non-trivial phase. Yet, we also find that ZBCPs are observed under different tuning parameters without simultaneous reopening of a bulk-gap. This demonstrates the importance of simultaneous probing of bulk and edge in the identification of Majorana edge-states.

[1] Braun Center for Submicron Research, Weizmann Institute of Science, Rehovot 76100, Israel. [2] Department of Condensed Matter Physics, Weizmann Institute of Science, Rehovot 76100, Israel. Correspondence and requests for materials should be addressed to M.H. (email: Moty.Heiblum@weizmann.ac.il)

The proposed prescription for the formation of a 1D topological superconducting phase[1–4] ignited a spree of experimental works[5–18]. With a 1D-like semiconducting nanowire (from now, wire) that bears spin–orbit interaction, applying a parallel (to the wire) Zeeman field can lead to spinless helical modes in a restricted range of chemical potential in the wire. Proximitizing the wire to a trivial (s-wave) superconductor is expected to induce a topological superconducting gap, accompanied by a pair of localized Majorana zero-energy modes at the two opposite ends of the wire. Consequently, tunneling electrons into the edge of the wire is expected to feature a zero-temperature quantized zero-bias conductance peak (ZBCP), $G_M = 2e^2/h$ (e—electron charge, h—Planck's constant)[19]. Indeed, ZBCPs had been observed in previous works with InAs and InSb 1D-like wires coupled to trivial superconductors[5–17]. A recent observation of a quantized value of $G_M$ was reported in ref. [18], for a review see refs. [20,21]. A ZBCP was also reported in tunneling experiments in ferromagnetic 1D atomic chains[22–24]. However, the necessary ingredients for inducing a topological superconductor, such as the presence of spinless helical modes and the ability to gate-control the chemical potential, were, to the best of our knowledge, not fully verified. Moreover, since a quantized ZBCP can also emerge at the edge of a 1D trivial superconductor due to a variety of mechanisms (e.g., refs. [25–32]), a direct and conclusive evidence of a Majorana edge state is still warranted. A strong and supporting evidence for that can be an observation of the reopening of the bulk gap with a simultaneous appearance of a ZBCP at the edge of the superconducting wire[33].

Kitaev provided a general procedure for the emergence of Majorana edge states in a topological 1D superconductor[1]. Such a 1D topological superconductor can be induced by coupling a trivial superconductor to a semiconducting 1D-like wire that harbors Rashba spin–orbit coupling accompanied by a parallel to the wire Zeeman field[3,4]. A sharp phase transition, which separates the topological and the trivial superconducting phases is expected to take place at $E_z = 2\sqrt{\Delta_{ind}^2 + \mu^2}$, with $\Delta_{ind}$ the superconducting gap in the wire, $\mu$ the chemical potential, $E_Z = g\mu_B B_Z$ the Zeeman splitting with the Landé g factor, and $\mu_B$ the Bohr magneton. The condition $\mu = 0$ is dubbed as the "sweet spot"—when the Fermi energy is located at the crossing energy of the two spin–orbit split energy dispersions[25,34–38]. A further increase in $E_z$ (to the above) reopens a (p-wave) topological gap. When the Fermi energy is within the Zeeman gap, the effective superconducting gap is the smallest of the two gaps: one at $\mathbf{k} = 0$ and the other at $\mathbf{k} = k_F$. At a small field, the dominant bulk gap is $E_g = 2\left|E_Z/2 - \sqrt{\Delta_{ind}^2 + \mu^2}\right|$ at $\mathbf{k} = 0$. At a high field, the dominant gap is at $\mathbf{k} = k_F$, decreasing with an increasing Zeeman field. The reopening of a topological gap must be accompanied by a zero-energy bound state at the end of the wire—the localized Majorana zero mode.

In the following, we describe our approach of simultaneous measurements of the energy-dependent tunneling density of states with a few tunneling probes placed at the bulk and at the edge of the proximized wire.

## Results

### Experimental setup
A scanning electron micrograph of the device and its schematic illustration are shown in Fig. 1a–c. Stacking fault-free round wurtzite InAs wires, with a 50–60 -nm diameter, were grown by the Au-assisted vapor–liquid–solid process in a molecular beam epitaxy (MBE) system on a (001) InAs substrate. For the wire growth, a very thin gold layer was evaporated on the InAs substrate after oxide blow-off in a separate vacuum chamber connected to the MBE system. After the

wires' growth, an ~7 -nm-thick Al layer was in situ evaporated on the side of the wires at a substrate temperature of −40 °C, creating a half-shell Al layer[38,39]. The wires were spread on a Si/SiO$_2$ substrate, already pre-covered by a wide metallic gate and 50 -nm-thick HfO$_2$ (grown by atomic layer deposition). Tunneling probes (TPs), 200 -nm wide, were made with an ~1 -nm-thick Al-oxide barrier and a top TiAu contact. A few TPs were placed at the bulk region and one TP at the edge of the 5–6 -μm-long proximized wire(s)[40–43]. The epitaxial Al was contacted by a TiAl superconducting contact (some 2.8 μm from the edge). As shown in Fig. 1c, the TPs couple only to the InAs wire (as the epitaxial Al was oxidized, thus preventing contacting it). It is worth stressing that in contrast to most pervious configurations[5–17], with existing regions along the wire not covered by the superconductor (thus allowing gate-induced stray field to penetrate these regions), our wires were lengthwise fully covered by the Al superconductor, thus mitigating an undesirable gate-induced potential. Supplementary Fig. 10 shows an example of the rich structure that the TP can detect.

Measurements were performed in a dilution refrigerator at 14 mK. The bulk and edge TPs were excited by $V_{AC} = 2$ μV RMS on top of a variable DC bias, $V_{DC}$. Each TP was excited by a different signal frequency, allowing two simultaneous measurements (with lock-in amplifiers), as shown in Fig. 1a. The induced superconducting gap in the wire was found to be close to that of the intrinsic Al film, $\Delta \sim 250$ μeV (Fig. 1d). A magnetic field with a magnitude of B was applied parallel to the wire, with a critical value of 1.6 T (placement accuracy was ± 2°).

### Transport measurements
Spectroscopy, in the form of tunneling conductance d$I$/d$V$ as a function of $V_{DC}$ and the Zeeman magnetic field B, is shown in Fig. 2. Measurements with the edge TP (TP1, tunneling resistance ~100 kΩ) and the two bulk TPs (TP2, 83 kΩ and TP3, 92 kΩ) were performed at a back-gate voltage $V_{BG} = -4.22$ V. This range of the tunnel resistance was found to be optimal (after multiple trials), as it allowed a measurable tunneling current while still minimizing field penetration into the wire. We believe that the potential fluctuations in the wire are stronger than the induced potential by the TPs. The two bulk TPs have a similar induced gap (which seems softer in TP2), while at the edge TP, it is slightly smaller—being at the edge of the wire.

### Bulk and edge behavior
Figure 2a shows the evolution of the tunneling current at TP1 (colored yellow in the inset). Two branches of an Andreev bound state merge at zero bias to form a single ZBCP at $B = 0.7$ T—persisting up to $B = 1.6$ T (when the Al superconductivity quenches). The ZBCP amplitude is $G_M = 0.22\ e^2/h$—likely limited by the finite temperature compared with the weak tunneling coupling and a finite dissipation[44]. The simultaneously measured evolution of the bulk gap in TP2 (positioned ~500 nm away from the edge, colored yellow in the inset) is shown in Fig. 2b. The gap closes at $B = 0.7$ T and reopens at $B = 1.05$ T—increasing to $E_g = 65$ μeV at $B = 1.4$ T (see also Supplementary Fig. 12). Notice that our resolution was ~25 μeV, thus limiting an exact identification of gap closure/opening. This also limits the ability to distinguish between a zero-bias energy peak and accidental finite energy states at the end of the wire with splitting that is below our resolution limit, insensitive to back-gate voltage. Moreover, the finite length of the wire may, in principle, restrict the expected closure of the bulk gap[45]—yet below our resolution limit. A similar behavior is observed with TP3 in Fig. 2c (placed 1200 nm from the edge, colored yellow in the inset). Here, the gap closes at $B = 0.95$ T and reopens to $E_g \sim 30$ μeV at $B = 1.4$ T. This difference is likely due to a long-range

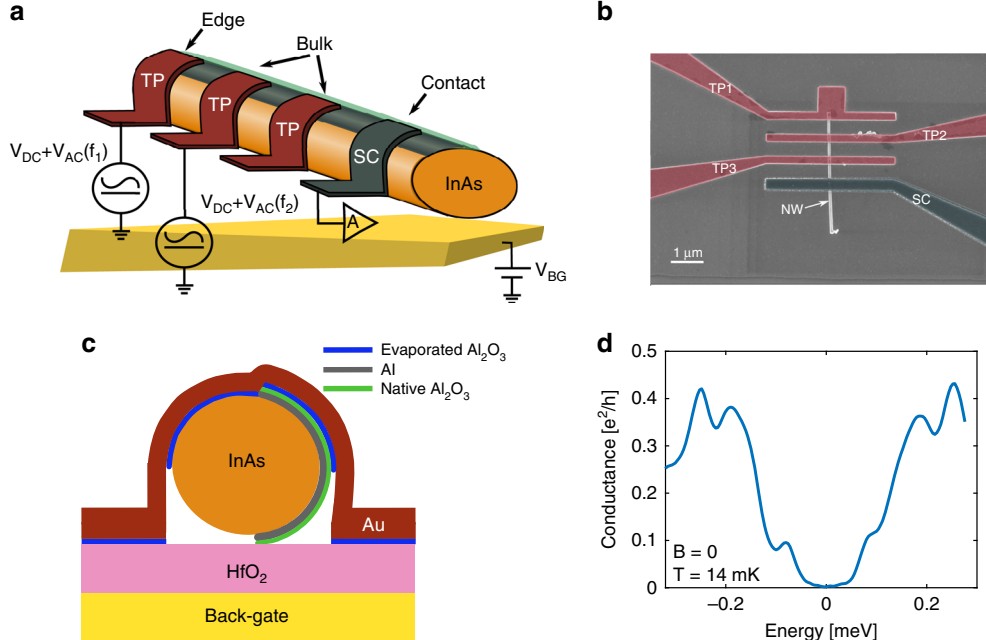

**Fig. 1** The nanowire structure and its nonlinear conductance. **a** Schematic illustration of the nanowire and tunnel probes (TPs). A superconducting Al contact, connected to the epitaxial Al on the wire (both in gray) is grounded via a current amplifier. Several TPs are connected to the edge and bulk of the wire (red), used as local voltage sources, separately or simultaneously. Each TP is only connected to the wire, since the epitaxial Al is isolated by a thick native Al oxide. The chemical potential is controlled via a global back gate. **b** An SEM micrograph of a typical device with three TPs (red) and a grounded superconducting contact (gray). **c** Schematic side view of a TP region: the InAs wire (orange) was covered by an epitaxial layer of epitaxial Al (7-nm thick, gray, on the right), which is oxidized (oxide thickness is 3 nm, green). The wire is placed on top of a metallic back gate, which is covered by a 50 -nm-thick $HfO_2$ insulating layer. Each TP is made of a thin Al-oxide barrier (blue) and a metallic Ti/Au contact (red). **d** The typical differential conductance measured by the bulk TP shows a hard gap with $\Delta_{ind} = 250\,\mu eV$ (at $B = 0$ and $T = 14\,mK$)

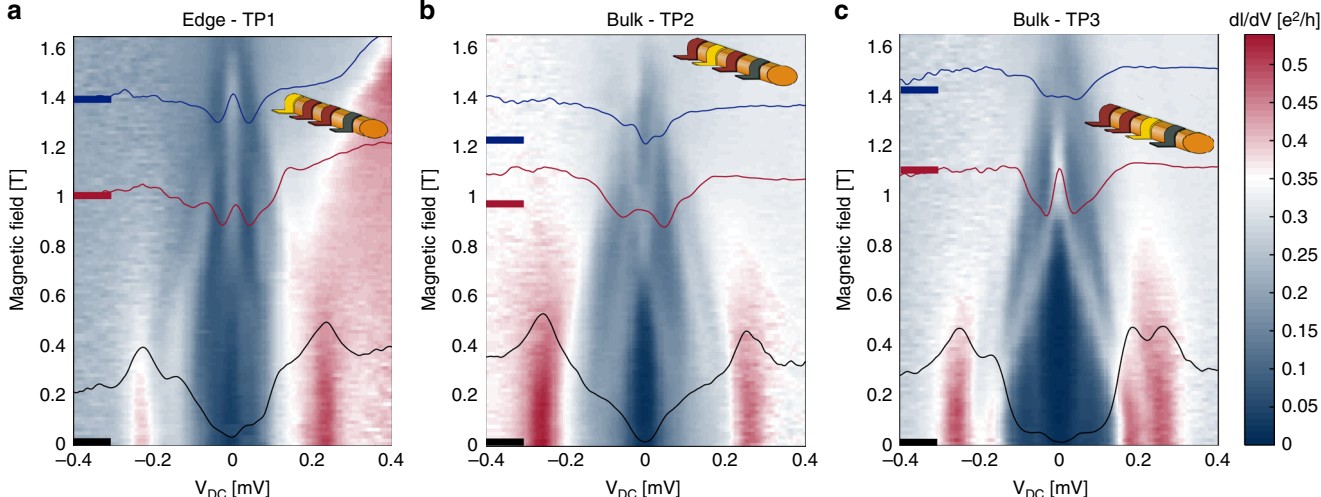

**Fig. 2** Conductance spectroscopy at the edge and at the bulk of the nanowire. Conductance as a function of the DC bias $V_{DC}$ and the magnetic field $B$, measured by three tunnel probes: **a** TP1—edge, **b** TP2—bulk, and **c** TP3—bulk. The active tunnel probe is colored in yellow in the inset of each plot. Measurements were performed on two TPs simultaneously (first TP1 and TP2, then TP1 and TP3), while the third TP kept floating. Cuts of the 3D plot at three different magnetic fields (marked by thick lines) are drawn on top (black, red and blue). **a** At the edge, a ZBCP appears at $B \sim 0.7\,T$ and persists until 1.6 T. **b** In the bulk, the trivial gap closes at $B \sim 0.7\,T$ and reopens (as a topological gap) at $B \sim 1.05\,T$, with $E_g = 65\,\mu eV$ at $B = 1.4\,T$. **c** The trivial gap closes at $B \sim 0.95\,T$ and reopens at $B \sim 1.4\,T$, with $E_g = 30\,\mu eV$. In all the measurements, the back-gate voltage is $V_{BG} = -4.22\,V$. The two bulk TPs have the same induced gap (though softer in TP2), while at the edge TP, it is slightly smaller. The color scale is common to the three plots

variation of the chemical potential along the wire—highlighting the difficulties in measuring nanowires[46–49]. Overall, the closure and reopening of the bulk gap take place for $B > 1.4\,T$ in all TPs. Supplementary Figs. 1 and 11 summarize the amplitude of the

ZBCP. It reaches a maximal height when the bulk gap (in TP2) is the largest. Supplementary Fig. 2 and Supplementary Fig. 3 show the behavior of a different device (#2) and thermally recycled data of this device, correspondingly.

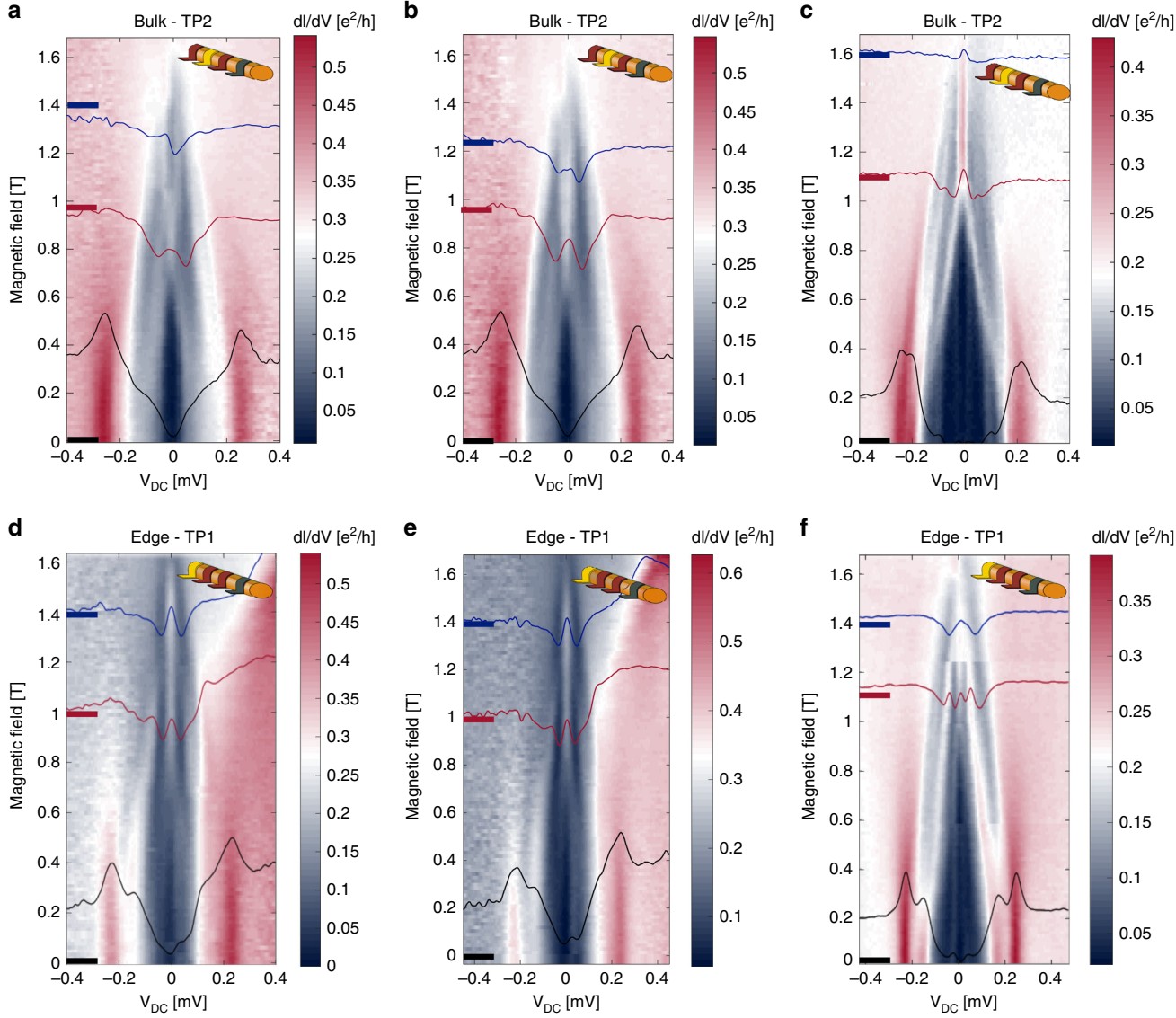

**Fig. 3** Correlation between bulk gap closing and the appearance of a ZBCP. Spectroscopy of the bulk (edge) as function of back-gate voltage (with TP2 (TP1)). The chemical potential is tuned by the back-gate voltage: $V_{BG} = -4.22$ V (**a, d**), $V_{BG} = -4.21$ V (**b, e**), and $V_{BG} = -4.13$ V (**c, f**). Cuts of the 3D plot at three different magnetic fields (marked by thick lines in left axes) are drawn on top (black, red, and blue). **a** The chemical potential is tuned to the "sweet spot" and the bulk gap reopens at $B \sim 1.05$ T, reaching $E_g \sim 65$ μeV at $B = 1.4$ T. **b** The chemical potential is slightly away from the "sweet spot", with a bulk gap $E_g \sim 25$ μeV. **c** Gap closing without reopening. In all three values of $V_{BG}$, the ZBCP appears in TP1 (see **d–f**); however, only in **a** and **b** the bulk gap reopens

**Gate dependence**. We studied the evolution of the gap in TP1 and TP2, as well as the ZBCP, as a function of the back-gate voltage (Fig. 3). The global gate voltage is not expected to affect the tunneling probability of the tunneling probes, since their coupling is determined by a high-barrier Al oxide. Hence, the observed dependence on the back-gate voltage reflects its effect on the chemical potential in the wire. Starting with Fig. 3a, where $V_{BG} = -4.22$ V (same as in Fig. 2), the bulk gap closes at $B = 0.7$ T and reopens at $B = 1.05$ T, to finally saturate at $E_g = 65$ μeV (see Supplementary Fig. 1). Being the largest observed gap, it is likely that the Fermi energy is placed very close to the "sweet spot" for the formation of a topologically nontrivial phase. Increasing the back-gate voltage slightly to $V_{BG} = -4.21$ V, the transition field moves up to $B = 1$ T to reopen at $B = 1.2$ T and saturates at $E_g = 25$ μeV (Fig. 3b)—see also Supplementary Figs. 13, 14. At $V_{BG} = -4.13$ V, the gap closes at $B = 1.1$ T and

never reopens in the full range of the magnetic field (Fig. 3c). In the latter case, a possible scenario (among others[50,51]) is that the chemical potential is too high; thus, an opening of a topologically nontrivial gap requires a Zeeman field that is near the critical field. As both TP1 (panel c) and TP2 (panel f, note the instability near $B \sim 1.25$ T) show similar traces of the ABS at low bias, we believe that these states exist throughout the entire wire. Note that at this chemical potential, $g^* = 3.9$, being the largest we measured in this device (assuming a simple model where the gap closes when the Zeeman splitting is equal to the superconducting gap).

A strong correlation between the appearance of the ZBCP and the reopening of the bulk gap is shown in Fig. 3a, d, as well as in Fig. 3b, e. Alternatively, as the bulk gap does not reopen in Fig. 3f, the ZBCP in Fig. 3c is likely a trivial ZBCP. Moreover, various scenarios of trivial behavior were measured, where a ZBCP appeared at the edge while the bulk did not show gap reopening

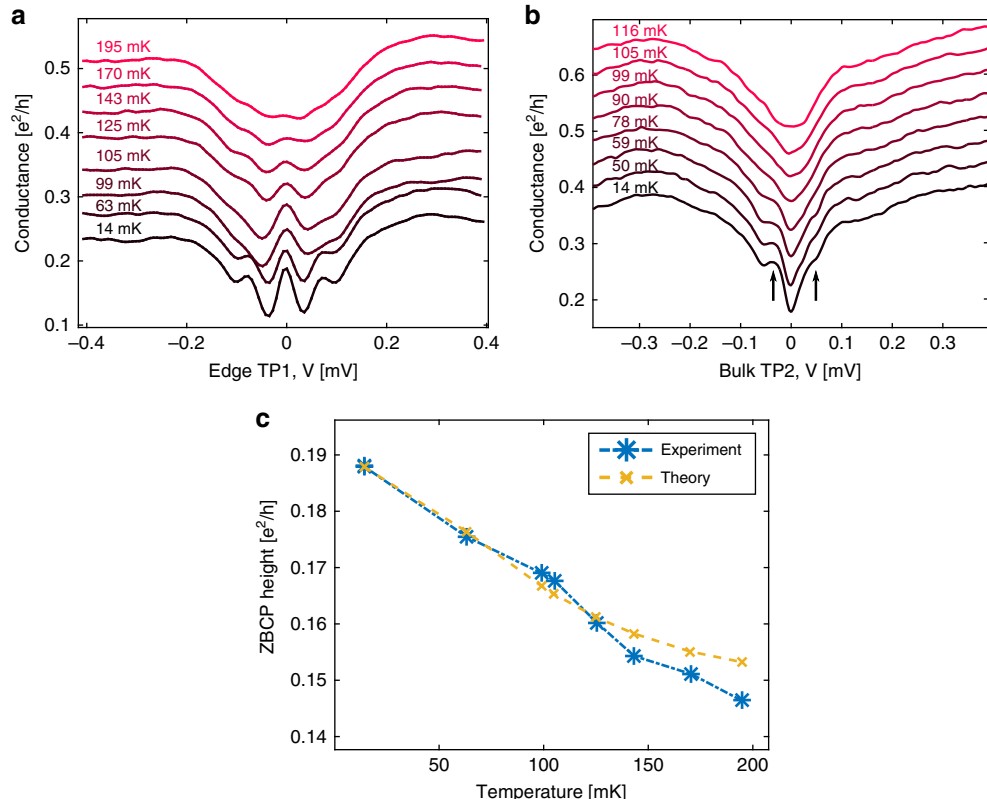

**Fig. 4** Temperature dependence of the ZBCP and the bulk gap. **a** Dependence of the ZBCP on temperature (barely visible at $T = \sim195$ mK). **b** Dependence of the bulk gap on temperature. At $T \sim 99$ mK, the superconducting gap shoulders disappear. Plots are shifted by $0.04 \times e^2/h$ for clarity. **c** The measured height of the ZBCP (blue) is plotted and is compared with a theoretical estimate (orange)—calculated by a convolution of the derivative of the Fermi distribution with the differential trace of the ZBCP at the base temperature (the temperature-independent broadening had been added)

(see Supplementary Figs. 4–9, 15). These findings clearly demonstrate the advantage of simultaneous probing of the bulk and the edge for identification of a topologically nontrivial state.

**Temperature dependence**. The temperature dependences of the ZBCP (Fig. 4a) and the bulk gap (Fig. 4b), at the "sweet spot" of the chemical potential and $B = 1.25$ T, were measured in the range $T = 14$–200 mK. At the base temperature, the full width at half maximum of the ZBCP was ~15 μeV, which is in reasonable agreement with the combined effect of thermal broadening ($3.5\,k_BT \sim 4.5$ μeV at 14 mK), the excitation voltage (5.6 μeV p–p), and the tunneling broadening. The evolution of the ZBCP is plotted as a function of temperature in Fig. 4c. The expected height was also calculated by a convolution of the derivative of the Fermi function and the conduction trace $dI/dV$ at base temperature (which already contained the temperature-independent broadening due to lifetime and excitation signal). The bulk gap starts at $E_g = 65$ μeV at $T = 14$ mK and softens with temperature (Fig. 4b). As expected, a complete gap softening, due to thermal broadening ($\sim3.5\,k_BT$), is observed around $T = 100$ mK.

## Discussion

The local density of states was measured along a fully proximitized InAs nanowire with a trivial superconductor (without bare regions, uncovered by a superconductor, along the wire). Tunneling probes were placed on the wire at the bulk and at the edge. Under a carefully tuned Zeeman field and the chemical potential, a correlation was observed between bulk-gap closure (of the trivial gap) followed by its reopening (as a topological gap), and the emergence of a zero-bias conductance peak (ZBCP) at the

edge. This is expected for the emergence of a localized Majorana state at the edge of the nanowire. The dependence of the bulk gap and the ZBCP on the back-gate voltage demonstrates the importance of correct tuning of the chemical potential in the bulk. Most importantly, we also find that for other values of the chemical potential and the magnetic field, a ZBCP can appear at the edge without evidence of a reopened gap in the bulk. This clearly underlines the importance of this technique and bears relevance to the observed ZBCPs. Our observations exclude disorder-induced short correlation length of the nanowire, which may separate the bulk probes from the edge probe, and the effect of a possible induced potential by the probes.

## Methods

**Sample fabrication**. For the deposition of the external superconducting contact to the Al thin layer on the wire, an argon ion-milling process was used for removing the Al oxide (using a beam voltage $V_B = 400$ V for 4 min), followed by in situ evaporation of a Ti/Al (5/80 nm) contact. The tunnel probes (TPs) were fabricated via gentle ion milling on the InAs wire for oxide cleaning (with beam voltage $V_B = 400$ V for 1 min), followed by the formation of an ~1-nm-thick Al-oxide barrier, which was formed in two steps: (1) evaporation of Al at a rate of 1.7 A s$^{-1}$ for 7 s. (2) Placing the sample in a load-lock chamber of the evaporator and introducing ozone at a pressure of $4.5 \times 10^2$ torr for 30 min. Afterward, the sample was returned into the main chamber for evaporation of the metallic contact of Ti/Au (5/80 nm). The TPs were 200–250-nm wide.

**Device important parameters approximate calculation**. We assume a single band in the wire with 1D density of

$$n = \frac{\sqrt{8m^*(\Delta_{so} + \Delta\mu)}}{\pi\hbar} \approx n_0\left(1 + \frac{1}{2}\frac{\Delta\mu}{\Delta_{so}}\right). \tag{1}$$

Here, $n$ is the electron density in the wire, $m^* = 0.022m_e$ is the effective band mass (with $m_e$ being the electron mass), $\Delta_{so} \sim 500$ μeV is the spin–orbit coupling energy

(the spin–orbit energy $\Delta_{so}$ is by definition the energy difference between the bottom of the band and the crossing point of the shifted parabolas in the presence of spin–orbit coupling), and $n_0 = \frac{\sqrt{8m^*\Delta_{so}}}{\pi\hbar} \approx 10$ electrons $\cdot$ $\mu$m$^{-1}$. The condition where $\Delta\mu = 0$ is referred to as the "sweet spot".

At the "sweet spot", the bulk gap

$$E_g = 2\left|\Delta E_Z/2 - \sqrt{\Delta_{ind}^2 + \Delta\mu^2}\right|, \tag{2}$$

closes at the lowest value of the Zeeman splitting, $\Delta E_z = g^*\mu_B B_Z$, being equal to the induced superconducting gap in the wire $\Delta_{ind}$. From Fig. 3a, corresponding to the "sweet spot" (with $V_{BG} = -4.22$ V), we find that the critical field for gap closure $B \approx 0.9$ T (this is an estimate, as the gap closes at $B \approx 0.7$ T and reopens at $B \approx 1.05$ T), with the Zeeman energy $\Delta E_z \approx 210\,\mu eV$. At $V_{BG} = -4.21$ V, the gap closes at $B \approx 1$ T, and thus the change in the chemical potential is given by

$$\Delta\mu = \sqrt{\frac{\Delta E_z(B=1\text{T})^2}{4} - \Delta_{ind}(B=1\text{T})^2} \approx 210\sqrt{\left(\frac{1}{.9}\right)^2 - 1} \approx 100\,\mu eV. \tag{3}$$

The levering factor is

$$\frac{\Delta V_{BG}}{\Delta\mu} = \frac{10,000}{100} = 100. \tag{4}$$

Hence, the change in the density of the electrons $\Delta n$, due to the change in the back-gate voltage of $\Delta V_{BG} \approx 10$ mV, is

$$\Delta n \approx n_0 \frac{1}{2}\frac{\Delta\mu}{\Delta_{so}} \approx 10\frac{100}{1000} \approx 1\,\text{electrons} \cdot \mu m^{-1}. \tag{5}$$

The estimated capacitance between the back gate and the wire is

$$C = \frac{e\,\Delta n}{\Delta V_{BG}} \approx 16\,\text{pF} \cdot \text{m}^{-1}. \tag{6}$$

We can also estimate the capacitance between the wire and the back gate by using the classical formula, where $h = 75$ nm is the distance of the center of the wire from the back gate, $r = 25$ nm is the radius of the wire, and $\varepsilon$ is the dielectric constant of HfO$_2$:

$$C_{classical} = \frac{2\pi\varepsilon\epsilon_0}{\cosh^{-1}\left(\frac{h}{r}\right)} \approx 300\,\text{pF} \cdot \text{m}^{-1}. \tag{7}$$

The order of magnitude discrepancy between the experimentally measured capacitance and the classical value is probably due to the crude simplicity of the classical model, as it does not include screening effects and the change in the position of the electron wave function inside the wire due to the presence of epitaxial Al. Also, the approximation of the classical capacitance assumes that the distance between the plane and the wire ($h$) is much bigger than $r$, which is incorrect for our case.

## Data availability

Data supporting the findings of this study are available within the article and its Supplementary Information files and from the corresponding authors upon reasonable request.

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

## Acknowledgements

We thank H. Inoue, A. Haim, Y. Ronen, Y. Cohen and Y. Reiner for useful discussions. We are grateful to J.-H. Kang and D. Mahalu for their professional contribution, and to Michael Fourmansky for professional technical assistance. We thank S. Das Sarma and J. Alicea for their useful comments on the paper. M.H. acknowledges the partial support of the Israeli Science Foundation (ISF), the Minerva foundation, and the European Research Council under the European Community's Seventh Framework Program (FP7/ 2007–2013)/ERC—Grant agreement 339070. H.S. acknowledges partial financial support of the Israeli Science Foundation (Grant no. 532/12 and Grant no. 3–6799), Israeli Ministry of Science (Grant no. 0321–4801 (16097)), and BSF Grant no. 2014098. H.S. is an incumbent of the Henry and Gertrude F. Rothschild Research Fellow Chair. Y.O. acknowledges support by the BSF and ISF grants and by the European Research Council under the European Community's Seventh Framework Program (FP7/2007–2013)/ERC —Grant agreement MUNATOP-340210 and under the European Union's Horizon 2020 research and innovation programme (grant agreement LEGOTOP No 788715).

## Author contributions

A.G. and E.B. contributed to this work in the sample design, device fabrication, measurement setup, and data acquisition. A.G. wrote the paper, made the figures, and did the data analysis. M.H. contributed to the sample design, data interpretation, and writing of the paper. Y.O. contributed to the data interpretation and writing of the paper. H.S. contributed to molecular beam epitaxy growth of the nanowires and writing of the paper.

## Additional information

**Competing interests:** The authors declare no competing interests.

