## [Peer Review File · Nature Communications]

Reviewers' comments:

Reviewer #1 (Remarks to the Author):

Since the Delft experiments in 2012, there has been a considerable debate of whether the observed zero-bias anomalies (ZBAs) in proximitized nanowires can be unambiguously interpreted as Majorana zero modes in the junction. Indeed, and although in many aspects the experimental observations are consistent with theory, the physical origin of ZBAs has been hotly debated since it is difficult to rule out alternative explanations (the main alternative mechanism being trivial Andreev bound states (ABS) mimicking Majoranas). Furthermore, the fact that the ZBAs are much smaller than the expected $2e^2/h$ is troublesome if one wants to invoke a topological origin. A second generation of devices from Copenhagen and Delft has resulted in much cleaner data with hard induced gaps and quantized conductance. The experiments presented here address another important question related to the fact that any ZBA of topological origin must be accompanied by a closing and reopening of the bulk gap, thus signaling a topological transition. Since the latest experiments from Copenhagen and Delft could be explained without invoking a bulk topological transition (see Vuik et al, arXiv:1806.02801 and Avila et al, arXiv:1807.04677), the experiments presented here have potential impact for the community and are thus worth to be considered for publication in Nature Comm. Now, the obvious question is whether these new results are convincing enough to settle the controversy about the observation of Majoranas. In my view, the authors only succeed partially in providing a convincing answer, as I argue below.

-First and foremost, the authors want to compare ZBAs measured with TP1 with bulk closing at TP2 and TP3. Surprisingly, they don't show clear cuts at $B=0$ to demonstrate that the spectral features of the sample (including of course the gap) are the same for TP1, TP2 and TP3. This is very relevant because they want to prove concomitance between ZBAs and gaps closing. The obvious question is therefore whether this "concomitance" does indeed correspond to the same physical situation. Such plots would also be important to identify the correct value of the gap in the proximitized wire (as compared to the Al gap). They write in the manuscript that the induced gap in the wire is close to the Al gap but this is not clearly seen in the contour plots (and clearly depends on TP, compare Fig 2a with Fig 2c at low B fields).

-Related to the previous point: the authors themselves acknowledge that TP3 corresponds to a different chemical potential configuration which, I think, prevents from using Fig 2c (TP3) in comparison with Fig 2a (TP1). More importantly, Fig. 3c clearly shows a robust ZBA after the gap closing, something unexpected in the bulk. Can the authors comment on this?

-Something similar occurs in Fig 3c where the bulk probe clearly shows a ZBA (what the authors call "absence of reopening"). This bulk measurement also shows traces of the ABSs present in fig 3f (almost parallel faint lines running at low bias). This seems to suggest that TP2 is not really a probe of the bulk only. Surprisingly, Fig 3f does not show a clear ZBA.

-The authors support their claims with a second device (extended Fig. 2). They claim that the same concomitant behavior is seen here but I only see a parity crossing of ABSs (the "closing gap" in Fig. b is just a couple of sub gap states well inside a bulk gap at around $\sim 0.2\text{mV}$). Furthermore, no clear ZBA is seen for TP1.

-The authors claim that their geometry avoids smooth potential/quantum dot effects. Yet, they show trivial ABS situations in the extended data (extended Fig.5) which are very similar to what one expects for such smooth/QD configurations. What is the origin of these ABSs? Can the authors comment on this?

-In the extended Fig. 7 they show a complex sub gap structure for bulk probes only. I am not sure if I understand the point that they try to make with such plots (apart from saying that all the spectral features and sub gap structure strongly depend on parameters).

-Concerning samples, the wires studied here are extremely long (around six microns) which makes it very difficult that transport is ballistic across the whole wire. Moreover, the distance between tunneling probes is non-negligible (more than half a micron according to the SEM micrograph). How do the authors know that measurements at different tunneling probe electrodes correspond to the same (=ballistic) physical system? Is disorder over such long distances not an issue here?

-Some minor points: when referring to "trivial" ABS explanations the authors only give credit to Das Sarma's group [26-29] which does not properly reflect contributions from other groups. These ideas date back to 2012 (Kells et al Phys. Rev. B 86, 100503(R), 2012 and Prada et al, 86, 180503(R), 2012). The reviews cited in the introduction are bit outdated concerning state-of-the art. Here, I would recommend to cite R. Lutchyn et al, Nature Review Materials, 3, 52, 2018; R. Aguado, La Rivista del Nuovo Cimento, 40, 523, 2017). The papers I mentioned at the beginning of the report (Vuik et al, arXiv:1806.02801 and Avila et al, arXiv:1807.04677) should also be cited.

In conclusion, this paper contains an interesting experimental setup that contains both edge and bulk tunneling probes in proximitized nanowires. This, in principle, allows to measure both the edge and bulk tunneling density of states (i. e Majoranas and gap closing). However, there are still important issues that need to be clarified in order to assess the importance and degree of novelty of these experiments, beyond incremental improvements, that would justify the claims in the paper and its publication in Nature Communications.

Reviewer #2 (Remarks to the Author):

The authors are presenting tunnelling measurements of the tunnelling current into both the edge and the bulk of a InAs nanowire proximitized by aluminum coating and in the presence of a magnetic field. Under these conditions such a wire has been predicted to become topological and support Majorana states at these ends. These states should appear as zero-bias peaks in the tunnelling conductance at the ends of the wire, for a magnetic field larger than a critical value. Their appearance should correlate with the closing of the bulk gap and reopening as a topological gap.

Some of the experimental observations reported here are consistent with this description, in that the bulk is closing and reopening at values of the magnetic field consistent with those for which the zero-bias peak (ZBCP) appears in the end-tunnelling conductance. However, there many instances in which the values for which the bulk gap is closing are different than those at which the ZBCP appears (a possible explanation being a variation of the chemical potential from the 'sweet spot'), and more troubling, for some parameters ZPCPs appear both in the bulk and end spectra, indicating that the origin of the zero-energy states observed is non-topological.

The observed results are very important, but I believe that the main message of this work should be that an observed edge ZBCP is clearly not a direct indication for a Majorana state and more investigations are necessary. This casts even more doubt about the origin of many previous observations of ZBCPs and their interpretation as Majorana. The reverse argument, i.e. the observation of a ZPCP correlating with a closing and reopening of the bulk gap implies that such peak correspond to a Majorana state, is however in my opinion consistent with the data but not fully supported by it as the only possible explanation, since it seems that one needs quite a lot of fine

tuning to find the right parameters to make it work and there is a lot of variation in the results. Since everything depends so much on the chemical potential, it is possible that one can find better alternative models and explanations that can describe well all the data, and that the correlation between the closing/opening of the gap with the formation of the ZBCP may as well be a coincidence with a mundane explanation, or it may well be the real thing.

However, I believe that the results presented here are very important and deserve publication since they show clearly that an end ZBCP may not necessarily be associated with a topological phase. Thus I would recommend publication provided that the message of the paper is made stronger in this direction.

Reviewer #3 (Remarks to the Author):

I read the manuscript 'Concomitant opening of a topological bulk-gap with an emerging Majorana edge-state' by Grivnin et al. with great interest and I congratulate the authors with this achievement, as I am well aware this research is extremely challenging to successfully perform. The principal investigators in this work are leading a research team which has been part of the 'initial wave' of Majorana experiments in semiconducting nanowires in the 2012 period, and to the best of my knowledge, this presents their first subsequent paper on this topic. Experimentally speaking, this research field is still very narrow, and as such, independent experimental work as reported here, is crucial for the field's long term viability.

To the best of my knowledge, this is the first experimental report on tunnelling spectroscopy performed both at the end and in the middle of a possible topologically non-trivial phase in a semiconducting nanowire. This makes the work highly original. Semiconducting nanowires are in my opinion the leading candidate for demonstrating non-abelian statistics in the future, and I am therefore convinced this research is very important. Lastly, although I have significant criticism on the completeness and presentation of the data, the experiment and its outcome is of very high quality. Those three aspects are my motivation why I would firmly support the publication of a future improved version of this work in Nature Communications.

Having said this, I find the current presentation of the work not publication ready, and surely not meeting the standards of Nature Communications. I therefore request significant improvements and modifications of the work. My main concern is the author's approach in their data selection to convincingly support the main conclusion of the work as expressed in their ambitious title. Clearly, they present (an) instance(s) in which the sought after behavior is observed. I therefore sympathize with the author's desire and attempt to make a grand claim with fitting title in this manuscript, this is how a research may get the right level of exposure. However, naturally, this also means the research will be measured against an extremely ambitious standard fitting such a claim. I expect this main claim to be unequivocally demonstrated in a coherent fashion throughout the work. Instead, the whole body of data under consideration here, spread over different devices, cooldowns, and gate voltage configurations, presents itself to me as a rich, kaleidoscopic picture full of nuances, rather than the simple picture imposed on it by the work's title, which covers just one aspect of the observations.

As I outline below, it is my strong suspicion this rich behavior is caused by some form of localisation and the formation of Andreev bound states underneath each tunnel probe, causing complicated behavior among the 3 tunnel probes. As this type of experiment has been suggested to experimentalist by theoreticians for years, and as such has long been awaited within the community, I do not find this complexity an objection against the work's scientific merit, in fact, I am impressed and intrigued by this richness in the data. However, rather than oversimplifying this for the shallow reader by placing it under the title's umbrella, I think the Majorana research community would in the long turn benefit much more from a more complete and direct presentation and discussion of the fact that the sought after behaviour of an apparent zero bias conductance peak (zbcP) at the wire's end

'concomitant' with the closing and reopening of a bulk gap is only one particular kind of behaviour found in this experiment, and it is very sensitive to gate voltage.

I am highly appreciative of the fact that I am only able to have this opinion because the authors show such a wide range of behaviours in the combined main text and extended data. However, I think that the authors should be less conservative in their approach, and address to some extent what I believe to be the elephant in the room – it is unfortunately not 100% clear at this stage if a topologically non-trivial phase with Majorana zero modes at its ends has ever been experimentally demonstrated, and this work doesn't change that. This is not to say that no progress has been made, in fact, the community as a whole has by now gained an impressive level of understanding of the device physics, as demonstrated again in the quality of the results under consideration here. To me, this double message is the main lesson of the past 6 years of research inspired by the initial wave of experiments, but it tends to get lost in triumphant one-liners such as this manuscript's title.

Provided the work's presentation improves significantly, I am convinced it deserves eventual publication in Nature Communication. The main aspects for the authors to focus on are:

1. a significant extension of the data included in the work, in particular, some basic device characterization data should be included in the extended data, the complementary gate dependencies to main text Figs 2 and 3 should be included in the main text, and whenever data on the main text device is shown, TP3's data should be included in cases where it has been omitted;
2. a critical revision of all references, I find there are important ones missing, and the function of others is unclear;
3. a thorough stylistic revision of text and figures.

In an attempt to aid the authors in this task, I have gone through their manuscript with great scrutiny, resulting in a long list of detailed comments and suggestions below. I expect the authors to respond to these in a satisfactory manner, and I genuinely hope this will result in a much improved, more balanced manuscript, which would benefit not only the authors but also the Majorana community as a whole, as I deem this work of significant importance to this research field.

Detailed comments (I will use the line number of the pdf document in my comments):

Title:

For your consideration and left to your discretion, although the word 'concomitant' is very nice and well chosen, please be aware that its interpretation may not be entirely obvious to a broad international readership largely consisting of non-native English speakers.

More generally, I am wondering at this stage what would be a 'decisive' dataset given the author's experimental set-up that would legitimate this title. I am of the opinion that the conclusion of the presence of 'a topological bulk-gap' demands a level of robustness of these observations (i.e. 'Concomitant opening of a topological bulk-gap with an emerging Majorana edge-state') in parameter space. The authors demonstrate this as a function of magnetic field strength, but I don't think they find a very convincing degree of robustness of this phenomenon in gate voltage as the closest experimental proxy to chemical potential. If I am wrong, please show relevant additional data and I would strongly recommend putting this in the main text. If I am right, we can obviously think of many reasons why the sought after observations are very sensitive to gate voltage (I will raise one such possibility), and yet still correspond to a topologically non-trivial phase, however, this is exactly the main claim of the title which I therefore expected to be demonstrated unequivocally without the need of all sorts of additional hypotheses. If the authors think I am wrong, please explain why. If the authors can think of another good experimental feature, beyond a wider extent in gate voltage range, that can demonstrate the main claim convincingly I'm happy to hear their reasoning and observe it in their data.

Abstract

Overall I feel the abstract is appropriate, and I only have some minor comments.

5: I find this opening sentence a bit over the top. In theory, these zero-modes are surely immune to

decoherence. In the experimental reality, it will be extremely difficult to protect these properly from nearby low energy states. Even this is achieved, still the problem of quasi-particle poisoning persists which will scramble up the computational basis provided by the majorana zero-modes. Although I surely think there is hope for engineering efforts to push this type of decoherence to long time scales, I find this sentence wrong and misleading, especially given this an experimental work.

6: is the comma after the word superconductor correct?

15: 'for particular gate-tuning' – my main worry about the experimental evidence provided to back the title's claim. I find it too particular. See also later comments regarding this point, please leave this comment here as it is a crucial aspect of this work.

18-21: It may be necessary to slightly expand on this, given my later suggestions.

Introduction

24: I don't like the referencing here. I feel given this is an experimental work, the authors only need to cite the absolutely essential theoretical papers necessary to support this sentence. To me, these are ref 1, 2, 9, and 10. Although ref 3 more or less arrives at the same results as ref 1 and appeared around the same time, it completely lacks the vision expressed in ref 1 which turned out to be the inspiration to the whole field, and it should therefore be omitted in my opinion. I feel the next crucial step was ref 2, where the insight was provided such systems can be engineered by combining other materials. Lastly, refs 9 and 10 give the actual 'recipe' implemented here. It may be useful to the interested reader to include some of reviews on this topic, such as ref 7, but it is not strictly necessary to backup this sentence.

25: Again I object to the referencing here. Clearly, the authors want to cite all experimental Majorana related works on semiconducting nanowires, which is to me indeed the right thing to do here. I find it particularly disturbing that the 2 papers by the De Franceschi research group are not cited here, nor anywhere else in the work. Those were very important experiments at the time, which made the whole community suddenly much more aware of the challenging nature of discriminating trivial zbcps from Majorana related zbcps. As this debate remains to some degree unresolved, they still serve that purpose and should be credited for that. Besides, some of the phenomena observed in the current manuscript may be explained by these mechanisms or closely related ones. Please include both works here, i.e. Lee et al. Nature Nanotechnology Vol 9 No 1 2014 and Lee et al. Phys. Rev. Lett. Vol 109 No 18 2012.

Furthermore, since all other experimental papers on Majorana fermions in nanowires are cited, surely the 2017 Science Advances paper by the Frolov research group should be included, Chen et al, Science Advances, Vol 3 No 9 2017.

32: the end of the sentence is in my opinion the appropriate place to cite ref 4.

32/33: please check language, I find the sentence somewhat inelegant. In terms of referencing, I suggest to not cite the two papers by Lee et al. at this point (please do include these in line 25), as they explicitly claim not to observe Majoranas. Please include the above mentioned paper by Chen et al. here as well.

36-38: I find using the phrase 'most crucial ingredients' arbitrary. They are indeed two essential ingredients, but I fail to see how these are more crucial than superconductivity if the whole idea falls apart with any of the 3 missing. Furthermore, although I agree with the spirit of the statement in the sense that independent verification of these aspect is lacking in the experiments, clearly attempts have been undertaken to show that these two ingredients are essential, namely by studying the zbcps behaviour while varying the magnetic field angle, or the gate potential.

39: I find the notion of 'nearly quantized' strange, I would not know how to define 'nearly quantized', nor how to discriminate it from 'not nearly quantized' or 'fully quantized'... I am raising this because this notion gives the false impression that refs 26-29 were highly original, which they aren't in my opinion - several important theory works predict trivial ZBCPs with a height of order G_0 and predate refs 26-29 by up to 5 years. Think of: Liu et al. Phys. Rev. Lett. 109.26 (2012): 267002; Bagrets and Altland Phys. Rev. Lett. 109, 227005, 2012; Pikulin et al. New Journal of Physics 14 2012; Mi et al.

Journal of Experimental and Theoretical Physics 119, 6, 1018 2015. Surely some/all these references should be included here, and one could consider cutting down on refs 26-29. Although refs 26-29 have some general relevance to this paper, a lot of the complicated behavior of ABS was covered earlier elsewhere already, in particular in works from Aguado and co-workers (e.g. ref 33, but there are some other relevant papers from those researchers), works from Loss and co-workers (e.g. ref 35), and even earlier works from Das Sarma and co-workers cover a lot of the physics reiterated in refs 26-29.

36-41: I find this passage as a whole somewhat weak. This research community would love to see a convincing experimental demonstration from a nanowire experiment of simultaneous tunnelling into the bulk and the end of a wire, confirming the Majorana hypothesis. Surely, a more compelling and pointy phrasing can be found to reflect this.

47: I got confused with the authors' definitions, please double check the following and point out the flaw in my reasoning or correct your own formulas and definitions: In the simplest model, the Majoranas originate from a pair of finite energy states, which (for $\mu = 0$) originate from the gap edges at $\pm\Delta$. Upon applying magnetic field, both states shift down with the Zeeman energy of a spin $\frac{1}{2}$ particle, i.e. $\pm \frac{1}{2} g \mu_B B$. That means to me the condition for the topological phase transition is reached when both states meet at the middle of the gap, i.e. when $\Delta = \frac{1}{2} g \mu_B B$, or for non-zero μ , when $\frac{1}{2} g \mu_B B = \sqrt{\Delta^2 + \mu^2}$. I would always define a Zeeman energy of a particle as the amount of energy it gains for a given magnetic field change. For spin $\frac{1}{2}$ particles, this is simply $\pm \frac{1}{2} g \mu_B B$. I find the current definition of E_z as twice that quantity confusing, an energy splitting is a difference between two energies and should incorporate some sort of delta symbol or so to stress that. Lastly, now both in line 47 and in line 53 an unnecessary factor 2 is incorporated.

48: spelling 'Bohr magneton'

50: spelling.

Experimental set-up

Generally I find this section functional and having the right amount of detail. I have questions about the following:

72: Why refs 38-40? It seems to me these references were chosen without much care, or at least, they are poorly connected to the sentence after which they are placed. I presume the authors want to refer to some earlier work from the semiconducting nanowire research community to show how their work fits in that bigger picture? Ref 38 is on coherent electronic transport in VLS grown silicon nanowires. I see only very little relevance of ref 38 to this particular experiment. After all, the authors are not providing an extensive review of the experimental progress in semiconducting nanowire research, but are rather describing their own set-up. Ref 39 is irrelevant in my opinion, as it is a) about top down fabricated nanowires, not on bottom-up grown nanowires as considered here, and b) its not a III-V nanowire. Ref 40 is somewhat relevant, as it is at least on transport through III-V nanowires. However, the InP tunnel barriers in those devices have no relevance to this experiment. In any case, these 3 citations report experiments geared towards exploring rather different physics, i.e. creating single electron quantum dots. Or are the authors in a subtle way implying this is relevant to their set-up? I agree Ref 41 is relevant. Other relevant references on the proximity effect in group III-V nanowires can be found from the works in the late 2000's period in the Delft and Lund groups and it may be appropriate to cite these here.

73: It is not a very pressing aspect to clarify, but nevertheless, can the authors clarify why they are sure there is no conductance between TP's and Aluminium shell? I.e. do you know the typical tunnel resistance between tunnel probe and epitaxial Aluminium? Is this a self-terminating oxide formed by exposing the wire to air after growth? If yes, I tend to agree with the statement, as such an oxide can be used as a gate oxide separating two Aluminium gates, and is able to withstand up to a few volts of potential difference, typically. Consider making this more explicit.

76-77, and also 90-93 in the 'Results' section: I find these very bold claims that are not supported by any theoretical or experimental evidence whatsoever.

My main concern and criticism towards this work is closely related to these statements. I would argue

the opposite: the tunnel probes are effectively at 0 voltage. InAs nanowires tend to have their Fermi level pinned to the conduction band edge at the nanowire surface, i.e. the bandgap is shifted towards negative gate potentials. I presume this is why a negative backgate voltage is applied in the experiments, to go to a more interesting regime of somewhat low density in the nanowire. I would think that given their (almost) 0 V potential the tunnel probes would tend to attract charge carriers. The backgate potential may result in the generation of weak potential barriers in between the tunnel probes. This may very well lead to the formation of (weakly) localised Andreev bound states, bound by the superconducting Aluminium gap and the backgate induced weak barriers. As such, my first intuition would be to think of this experiment as 3 parallel 'Andreev quantum dots' which share the same common superconducting lead. I expect such an effect to vastly surpass the influence of any potential fluctuations due to defects within the wire or impurities on its surface. Obviously, my concern is of great impact on the interpretations of the results reported here, as in case of localized Andreev bound states underneath each TP, it is not very surprising that rich behavior is found upon comparing measurements from different TP's. It is not at all clear in such a scenario that a single, undisturbed topologically non-trivial phase is present in the device.

Can the authors please provide compelling experimental evidence why the picture I'm sketching is incorrect? If they can't, can they please remove such claims, and take my suggested picture (or some related variety), into careful consideration and rewrite the article from this perspective?

Many of the requests for additional data that will follow are closely related to this aspect.

79-82: This sentence implies that simultaneous measurements were performed on several tunnel barriers, and that this is depicted in Fig 1a. I see only a single voltage bias source in that picture. More generally, is this how you did your experiments? Or did you first perform an experiment on the edge TP with bulk TP left open, and then repeat it with the bulk TP leaving at TP open (or vice versa)? If you did a true simultaneous measurement of several TP's, shouldn't you place current meters in all branches of the circuit, i.e. on the input side of each TP, and as a sanity check on the (common) drain side as well? Either way, this statement or Fig 1a is incorrect. I think it is important that you make crystal clear everywhere in the manuscript which TP's were biased and measured, and which ones were presumably left floating.

83: spelling, critical magnetic field I presume.

Figure 1 + caption

Fig 1a - see comments above on measurement circuitry. I suspect your nanowires are hexagonal in cross sectional shape? If yes it would be better to depict it like that, a small effort to make the figure much more realistic.

Fig 1b - label TP's, contact, nanowire and include materials.

Fig 1c - I find this figure very unclear. There are many layers, many colors and most of them are left unlabelled. Again, if your wire has a hexagonal cross section please depict this. And are these large voids realistic? Please improve.

Fig 1d - Consider placing $B = 0$ T in the panel to make this very clear.

287-288: fix language, incomplete sentence.

295: Can you plot this on a log scale and add this to the extended data, to make clear there is indeed 2 or 3 of magnitude suppression between $V_{\text{bias}} = 0$ mV and $V_{\text{bias}} = \pm 0.3$ mV? If this is what you find, I would agree with this statement. Otherwise, data from a high resistive tunnel probe would be good to show in the extended data (e.g. a line trace both in linear and log scale at $B = 0$ T taken on the device on which extended data Fig 7 is taken).

Results

I already expressed my general concerns regarding the interpretation of these data above. Here my detailed comments.

90-92: I do not understand this claim. In fact, I would think that the more opaque tunnel probe of the device shown in extended data Fig 7 was better, as its sharper nature clearly resulted in better resolution of the rich subgap features present. Also, I believe the tunneling conductance through a possible Majorana zero mode should scale with tunneling probability T , whereas for general Andreev

bound states this rather scales with T^2 , i.e. measuring using more opaque barriers could be beneficial in suppressing tunneling into trivial states (obviously, it will suppress the zero bias peak height). Lastly, I find the comment '...while minimizing field penetration in the wire.' rather cryptical. Please explain.

92-93: see above.

96: language, some sort of verb seems missing '...to form a single ZBCP...' sounds better to me

101-102: No experiment can determine an exact gap closure / opening, it is always limited by the measurement resolution (also in B field stepsize, by the way). I suggest instead of all the tildes in the preceding sentences, write $B = 0.7 \pm ? T$ etc., and when you give a gap size at a certain field just write $E_g = 65 \pm ? \text{ microeV}$ at $B = 1.4T$. Seems more exact to me than the tildes.

102-104: I do not really get this comment and its relation to ref 45. Please clarify.

106-108: Agreed. I would go much further and argue that it is not obvious that what is measured at TP1, TP2 and TP3 is necessarily strongly correlated (beyond the obvious correlations given by sharing the same bulk gap), for above mentioned reasons of TP induced potential fluctuations.

108-109: For your consideration: this is in my opinion not a very important point, as one can think of many reasons why the ZBCP may fluctuate in height as a function of global system parameters such as magnetic field. Also, although Extended Data Fig 1a appears convincing, it is somewhat deceptive, because around 1.4 T, already the bulk superconducting gap starts to be suppressed significantly. I think it is difficult to conclusively discriminate in a quantitative manner the increase in height of the ZBCP from an overall increase in conductance within the superconducting gap.

Figure 2 + caption

Consider placing something like 'Edge', 'Bulk - 1', 'Bulk - 2' above panels a, b, c. The little insets with device schematics are useful, but from just looking at the figures and schematic it may not be fully obvious that 'yellow' means this is the TP used in that scan. In panel c I find the dotted guides to the eye and vertical arrows obscuring the gap closure and reopening. Lastly, I strongly suggest to make plots similar to those shown in Extended data Fig 3 c and d (taking my comments on those into account) and place these in the extended data.

301: make clear what happens to the other two TP's when the other one is biased, i.e. presumably they are left floating?

303: its marked by short tick lines, not arrows.

Extended Data Figure 1 + caption

See my comment on line 108-109. I see no harm in leaving this Figure in the Extended Data, but find its value somewhat limited. Also, the division in a green trivial and blue topological section depends heavily on the interpretation of this data. The authors could consider choosing a more neutral phrasing (although I agree some sort of partitioning of the behavior as a function of B is useful). Please explicitly mention this is (presumably) based on the the exact same dataset as plot in Fig. 2a and b, i.e. explicitly state panel a is a linecut at zero bias of Fig 2a and panel b is calculated based on the data of Fig 2b.

Results - continued

110: I kindly request the authors to show additional data. I find the amount of data presented in this part of the study is insufficient to support many of the conclusions made in this work. I would like to see the following data:

1. Bias dependent tunneling spectroscopy vs Vgate for all 3 TP's, at a range of B fields starting from 0 T, with at least a few magnetic fields before the supposed topological phase transition, and a sufficient density of data right before, during and after this possible phase transition. Such data should focus on the low energy behavior, i.e. the subgap states, so a bias range of $\sim \pm 0.5 \text{ mV}$ clearly suffices. The gate range should surely include the whole range covered in Fig 2, 3 and Extended data Fig. 5 and 6. If such data is not available, I am somewhat sceptical about the overall scientific merit of the work, and my final recommendation for acceptance or rejection in Nature Communications will largely depend on the completeness of these data. Please anticipate a request from my side for inclusion of such data in main text and/or extended data.

2. Bias dependent tunneling spectroscopy vs V_{gate} for all 3 TP's with a specific focus on ruling out / confirming localisation effects. I find this manuscript lacks a set of basic characterisation data of the device's behavior as a function of gate. I think this should be part of the Extended Data, and surely shared with me given my concerns. I am thinking of a large scale conductance map at $B = 0 \text{ T}$, ranging in gate space from pinch off (can you actually pinch off your devices? If not, I expect repetitive or stable behavior below some negative gate voltage?), until saturation at high gate voltage, with a sufficient bias voltage range, i.e. order 5 mV. I would like to see such a dataset for all 3 TP's.

3. Bias dependent tunneling spectroscopy vs magnetic field - the main text device has 3 TP's, and I would like to see the equivalent data of Fig 3 (now shown only for TP1 and TP2) also for TP3. Already Fig 2c shows that non-trivial difference between TP 2 and TP3 are present at fixed gate voltage, and I think Fig 3 should simply be expanded from 6 till 9 panels, including TP3 (unless the data is somewhat 'boring', in which case this should still be shown as Extended Data).

110-113: Unnecessary in my opinion, I am in agreement, but you provide no argument why this holds and the sentence isn't very exact. I don't think anybody will really doubt your backgate is having a very significant effect - any conductance line trace at fixed V_{bias} while varying V_{gate} will show huge variations, proving this point. Mind you that in my opinion you cannot rule out that tunnel strength may fluctuate as a function of V_{gate} - see comment on line 345-346 of Methods section below, however, this is a different point.

116: 'Being the largest observed gap...' and 119: '...and never reopens in the full range of magnetic field (Fig. 3c).': typical statements that needs backing up by additional data as requested under 1) above. You have the opportunity here to demonstrate the extend in V_{gate} of the ZBCP and reopened bulk gap, I strongly recommend doing so.

121-122: Why do you have fluctuations in your effective g-factor? I am a bit sceptical about assigning effective g-factors to these states moving in B field. Clearly, they originate from a quasi-continuum at finite energy, where level repulsion causes a much 'slower' dispersion in B field. Often 'along the way' to zero energy other levels also disperse in B-field, which may lead to further distortions from a simple linear B field dependence. Lastly, how long a level 'sticks' to zero bias in magnetic field also heavily depends on repulsion from the nearby finite energy states. This influences assigning effective g-factors and points of 'gap closure' and 'gap reopening', and answering the naively binary question if a ZBCP is 'robust' or simply a cause of trivial ABS 'sticking' to zero bias for a while. Also, the coupling strength to the superconductor heavily influences all these behaviors. Ref 48 is quite illuminating in this respect and illustrates many of the points I am raising, you could consider citing this work in a more prominent spot in the text.

122-123: Another statement that needs extensive back-up by providing gate dependent data as requested under 1) above.

124-125: I appreciate the authors showing this behavior, as it constitutes a 'negative' result in the context of this paper's main claim. Naturally, the question arises why any of the ZBCP's behavior is strongly correlated with that of the 'bulk gap' measured at TP2. This connects to my overall comment of the possibility of probing individually localised states underneath each TP. A priori, this doesn't necessarily mean all the reported behaviors of ZBCP and bulk gap are topologically trivial, but obviously a lot of nuancing seems very appropriated.

Figure 3 + caption

Please include data for TP3, unless it is identical to TP2 (Fig 2 doesn't suggest so), in which case it should still be included as part of the Extended Data in my opinion. I suggest to follow the same ordering of panels as in Fig 2, i.e. each row corresponds to the same gate voltage. That makes it much easier to compare behavior at different TP's for constant field. This would make Figure 3 a 9 panel figure. The authors could also consider combining Fig 2 and Fig 3 in a single Figure 2, and add a new Figure 3 with the complementary data of varying V_{gate} at constant B field. That would in my opinion be a much more comprehensive and consistent dataset.

As commented with Fig 2, consider placing something like 'Edge', 'Bulk - 1', 'Bulk - 2' above panels. I find several of the dotted 'guide-to-the-eye' lines obscuring nuances in the data, in particular in panel

a around the crossing point of the states, for panel b around 1.3 T, and for panel f around 1.25T
A detail: In panel f, a switch is visible around 1.25T. Please share your opinion on the origin of this, and the occasional other switch visible throughout the data; is it an electron trap in the oxide (dis)charging, and if yes, it seems somewhat counterintuitive this happens as a function of magnetic field? How do you ensure such switches do not affect conclusions related to changing the chemical potential in the wire? Consider placing a short comment somewhere in the caption / main text/ methods section about the occasional occurrence of such switches (I congratulate the authors on achieving the level of stability displayed in the data throughout the paper, charge(?) switches seem rare which is great).

Lastly, I recommend to provide plots similar to those shown in Extended data Fig 3 c and d (taking my comments on those into account) and place these in the extended data.

Results - continued

127-137: Although important from a viewpoint of completeness, I think this section and Figure 4 belong in the Extended Data. It is by now well established that the temperature dependence of the ZBCP does not aid in uncovering its origin, and follows a trivial pattern of suppression as a function of T. Similarly, the 'bulk gap' closure appear rather trivial to me. I encourage the authors to focus instead on the in my opinion much more important and interesting questions of gate dependence and comparing all the different types of behavior observed within the main device, and across all devices. Figure 4 + caption

See comment on lines 127-137 above. Panel a: There is still a tiny ZBCP left at 195 mK. Could you show a few more traces at higher temperatures to prove the ZBCP really disappears? Panel b: I find it somewhat strange you don't show curves up to a similar T value as in panel a. Naturally, one now wonders if anything happens between the highest T value of 116 mK shown in panel b and the value of 195 mK shown in panel a. Lastly, it could be beneficial to duplicate the data as a colorplot, as different types of plotting draw attention to different types of features.

Summary

144-145: language, ZBCP is already defined, not sure what journal policy recommends here. 'This is expected for the emergence...' seems better.

147-151: I feel this should be slightly expanded and rewritten based on the revisions of the paper I am suggesting.

Methods

Sample fabrication

This section is good as it is, except from one comment.

345-346: The effective tunnel barrier strength is not just dependent on the tunneling through the oxide (I indeed agree there's very little reason to expect such a tunnel strength will depend on the backgate voltage), but also depends on the exact probability density of the orbital nanowire wavefunction(s) near the tunnel barrier, which obviously will depend on the backgate voltage (although the effective OV potential on the TP's may keep things fairly stable right underneath the TP's). In view of this I find this statement a bit misleading, it is anyway not necessary in this methods section.

Device parameters

354-355: If my earlier reasoning is correct, I would prefer defining E_z without the factor 2 in there.

349-361: I find this estimate and the method used reasonable, and the lever arm found fits well with earlier experimental work on nanowire quantum dots (typically lever arm is of order 10% vs 1% in this work), especially considering the extra screening due the aluminum shell.

363: The authors find a change in charge of the order of 1 electron per micrometer of nanowire per 10 mV change in backgate. This again emphasises my earlier point of the possibility of localisation effects being very important in this work: your TP's have a width of ~0.2 microns, i.e. pretty much of the order of 1 micron. This compares quite favorably to the number you estimate (factor 5 smaller), given the many assumptions that goes into the estimate.

365-373: I think this is better left out. By now it is well established that classical capacitance

calculations (miserably) fail in these systems and the appropriate approach is to self-consistently solve the Schrodinger-Poisson equation. See Vuik et al., New Journal of Physics, Volume 18, March 2016 and consider citing this paper, as it is relevant to this work.

Extended data

Fig 1: see comments above.

Fig 2: Consider labelling panels with 'edge' and 'bulk'. Can you extend the plot by another ~200 mT? It seems the gap hasn't fully closed yet. Panel a: I find the 'guide-to-the-eye lines somewhat obscuring the details. Given that the tunnel barrier is so transparent, are you even sure there is a ZBCP? Maybe it is just two levels coming very close and appearing as a broad ZBCP? Lastly, I recommend also plotting the linetraces for this data.

402-404: This is not a very strong statement, first you write that a ZBCP appears at 0.5 T, then that the bulk gap reopens at 0.6 T. I don't see a problem there? Having said that, I do indeed agree you may very well have had a different chemical potential for both tunnel probes.

More generally, is there more data on device 2, i.e. does it display other types of behavior for different gate voltages? It could be worth extending the figure with a few other types of behavior, if measured.

Fig 3: Consider labelling panels a and b with 'edge' and 'bulk'. Panels c and d: I find the lines unclear because of all the individual dots. I am quite fine with just plotting a narrow line without individual datapoints. Also, I find the many colors not necessarily helpful, you could consider plotting it all in black and possibly indicate a few important traces by using a different color, such as onset of the ZBCP or point of gap closure / reopening.

417-421: Rather grand statement given the quality of the data. I fail to see: 1) why the ZBCP in panel a needs to be Majorana related, all I see is a gap closure pushing 2 pairs of ABS together to form a ZBCP right before superconductivity disappears. 2) a reopening of the gap in panel b, same arguments as for panel a). I find this a too large stretch. I would leave this dataset in the Extended Data, but give a more realistic down-to-earth description.

Same as for Fig 2 on device 2, is there more data that display other types of behavior for different gate voltages? It could be worth extending the figure with a few other types of behavior, if measured.

Fig 4: Consider labelling panels a and b with 'edge' and 'bulk'.

Panel a): It seems a second pair of ABS 'merges' with the ZBCP at 0.9 T? Above that field value, the ZBCP is much broader, maybe there is a low energy 4 fold split conductance peak that gets washed out by the tunnel broadening?

panel b): Isn't there a weak ZBCP present at 1.2 T? Please comment.

Again, is there more data on device 1 after thermal cycling it, i.e. does it display other types of behavior for different gate voltages? Could you find a 'gap closure' and ZBCP appearance at similar fields as reported in Fig 2 and 3 main text? I recommend also plotting the linetraces for this data.

Figs 2-4 form a useful additional dataset showing reproduction of the main features. In fact, I wouldn't mind extending the data in all 3 cases somewhat to cover a broader type of behaviors across the 3 devices.

Fig 5: I think this is an important dataset, that should be connected in some way to the results in the main text Figs 2 and 3. By changing the backgate voltage by a very small amount of 0.07V and 0.08V compared to the results in main text panels 3a,d and 3b,e respectively, suddenly this ZBCP at the edge became trivial. This really asks for some clarification. Is this the same feature as the one in the main text which is claimed to be non-trivial? If not, please proof this with experimental data, for example a continuous V_{gate} vs V_{bias} scan covering the whole range between main text Figs 2,3 and this figure, tracing the ZBCP. If it is the same feature, the authors should combine these data with those shown in the main text and make a single main text figure out of it. If it is not the same feature, I still strongly recommend to move this into the main text, as part of a separate figure that shows 'trivial' ZBCP's, with a corresponding discussion.

Some more details: I think the data for TP3 should be included here as well (as with main text Fig 3). Label panels with 'edge', 'bulk 1', etc. Lastly typo in line 448, 'below' is what it should read.

Fig 6: Again, an important dataset. How does this connect (from an experimental viewpoint) to main

text Figs 2 and 3, and the previous extended data Fig. 5? Do you have more data from neighbouring gate values? If you can prove panel a) is a distinct feature from the one shown in main text Figs 2 and 3, it may still be worth showing these results in the main text as a separate figure. Isn't this another indication that what you measure at each TP isn't strongly related? Could you please include TP3's data as well?

Fig 7: I find this a very beautiful dataset, it should surely stay in the Extended data. Please consider my earlier remark, I don't really understand why you think this more opaque tunnel barrier is not suitable for edge tunneling spectroscopy. Please check the grammar in the caption there are several errors.

Editor

We suggest that this first paragraph should be also sent to all three Referees.

We thank the Referees for their comments and the time they invested in the reviews. We hope we've clarified most of the issues, and emphasized our main message – the importance of simultaneous bulk and edge measurement. We have been working with nanowires for some time and recognized (like many experimentalists do) the difficulties with working with them: their uncontrolled disorder, their 'switching' behavior, the non-reproducibility from wire to wire, and the difficult in controlling the chemical potential. The work we describe in this paper was a two-year effort of Ph.D. and M.Sc. students, who fabricated and tested many wires. The paper presents the best we can offer. As the paper may not be ideal, it may serve as beacon –being the first type of its kind - to similar (and maybe better) future works. This was also the case with the first nanowire/Majorana papers – being far from perfect, they ignited the candle that still burns.

Referee 1

You find below your full report with our detailed comments attached (in *italic blue*).

First and foremost, the authors want to compare ZBAs measured with TP1 with bulk closing at TP2 and TP3. Surprisingly, they don't show clear cuts at $B=0$ to demonstrate that the spectral features of the sample (including of course the gap) are the same for TP1, TP2 and TP3. This is very relevant because they want to prove concomitance between ZBAs and gaps closing. The obvious question is therefore whether this "concomitance" does indeed correspond to the same physical situation. Such plots would also be important to identify the correct value of the gap in the proximitized wire (as compared to the Al gap). They write in the manuscript that the induced gap in the wire is close to the Al gap but this is not clearly seen in the contour plots (and clearly depends on TP, compare Fig 2a with Fig 2c at low B fields).

Line cuts at $B=0$ of TP1, TP2 and TP3 were added to Fig. 2 (in black), and also plotted in the figure below. Similar features appear in all three tunnel probes (marked by Δ , \square , \circ). The different tunnel probes show these features at slightly different energies and conductance values.

Note that the softness of the features and the overall conductance are very sensitive to the TP's resistance. The three TPs have slightly different resistances, which in turn result in a slightly different tunneling conductance values and appearance.

The two bulk-TPs have the same induced gap, while in the edge-TP it is slightly smaller. This difference results from the fact that TP1 sits at the edge of the wires – making the induced gap smaller - while the other are in the bulk. We see such a behavior in other devices too. For example, device #2 where the induced gap at the edge is $227 \pm 5 \mu\text{eV}$ and at the bulk it is $243 \pm 5 \mu\text{eV}$.

Related to the previous point: the authors themselves acknowledge that TP3 corresponds to a different chemical potential configuration which, I think, prevents from using Fig 2c (TP3) in comparison with Fig 2a (TP1). More importantly, Fig. 3c clearly shows a robust ZBA after the gap closing, something unexpected in the bulk. Can the authors comment on this?

The goal of Fig. 2 is to present the tunneling spectroscopy in different locations in the wire and thus point out the effect of unavoidable disorder. TP3 shows (in general) similar behavior to that of TP2 with a small shift in the magnetic field where the gap closes and reopens – such a shift reflects the slightly different chemical potential. Yet, the overall transition to a topological regime (everywhere) takes place for $B > 1.4T$.

In Fig. 3c we observe a gap closure without reopening as the chemical potential increases. We believe that in this configuration the chemical potential is higher than

in Fig. 3a and 3b; hence, re-opening of a topological gap requires a Zeeman field that is near the critical field. Note that our energy resolution is finite, being $\sim 25\mu\text{eV}$, and a very small gap may appear as a conductance peak.

Something similar occurs in Fig 3c where the bulk probe clearly shows a ZBA (what the authors call “absence of reopening”). This bulk measurement also shows traces of the ABSs present in fig 3f (almost parallel faint lines running at low bias). This seems to suggest that TP2 is not really a probe of the bulk only. Surprisingly, Fig 3f does not show a clear ZBA.

Figures 3c and 3f present a configuration in which a ZBCP appears at the edge of the sample but the bulk-gap is effectively zero (being smaller than our resolution). Figure 3f was indeed not clear enough because of a small jump that appears around $B=1.25T$. Line-cuts were added to the plot and show a clear ZBCP at the edge of the wire.

As noted correctly, both TP1 and TP2 show similar traces of ABS at low bias. We believe that these ABS exist throughout the entire wire, edge and bulk, thus seen in the tunnel probes. This emphasizes our main claim that observing “sticking of Andreev states” at the edge of the NW that leads to a robust ZBCP do not provide sufficient evidence for a Majorana ZBCP.

The authors support their claims with a second device (extended Fig. 2). They claim that the same concomitant behavior is seen here but I only see a parity crossing of ABSs (the “closing gap” in Fig. b is just a couple of sub gap states well inside a bulk gap at around $\sim 0.2\text{mV}$). Furthermore, no clear ZBA is seen for TP1.

*We thank the Referee for this comment. We have reexamining the data and add cuts for clarity, we think that the data may be interpreted by stabilization of the topological state at $B>0.7T$, as for that value there is a ZBCP at the edge and not in the bulk. However, the behavior at low field demand further interpretation, as it can result due to the high transparency of the edge-TP (10 kOhm) and the short coherence length. However, following the Referee’s remark we agree that the evidence for topological state are not clear enough and therefore decide to remove the figure (see figure below) from the **Extended Data** in order to avoid confusion (as one can see we added a large number of figures to the Extended Data section).*

The authors claim that their geometry avoids smooth potential/quantum dot effects. Yet, they show trivial ABS situations in the extended data (extended Fig.5) which are very similar to what one expects for such smooth/QD configurations. What is the origin of these ABSs? Can the authors comment on this?

Our devices are covered with in situ grown Al SC along half the perimeter of the wire, flooding the wire with carriers and screening local potential fluctuations (no bare wire is exposed). This is the reason we believe a QD configuration is less likely at the end of the wire due to a density depletion resulting from the global gate – as natural in customary fabricated devices. While less likely, we show an ABS appearing due to a local potential minimum due to disorder. This intends to show how important is to perform correlated measurements.

In the extended Fig. 7 they show a complex sub gap structure for bulk probes only. I am not sure if I understand the point that they try to make with such plots (apart from saying that all the spectral features and sub gap structure strongly depend on parameters).

The goal of Fig. 7 (Fig. 10 in the revised version of the Extended Data) is to show the potential spectroscopic abilities of such tunneling probes. Here, we wanted to show that these tunnel probes allow in general a very clear and sharp measurement of the DOS in the nanowire.

Concerning samples, the wires studied here are extremely long (around six microns) which makes it very difficult that transport is ballistic across the whole wire. Moreover,

the distance between tunneling probes is non-negligible (more than half a micron according to the SEM micrograph). How do the authors know that measurements at different tunneling probe electrodes correspond to the same (=ballistic) physical system? Is disorder over such long distances not an issue here?

The length of the entire wire is indeed long and we don't assume the transport is ballistic all along. Our goal was to measure transport at different positions on the wire and estimate the disorder length. Indeed, we show a much better correlation between edge-TP1 and the nearby bulk-TP2, and a shifted chemical potential at the further bulk-TP3. The overall length between TP1 and TP2 is $\sim 0.5\mu\text{m}$ - being similar to other 'Majorana type experiments' (e.g. Zhang et al. 2018). It's important to note that the nearest bulk-TP should be far enough from the edge so that there will be no overlap of the edge Majorana state (and collapsed gap) with that of the bulk's gap.

Some minor points: when referring to "trivial" ABS explanations the authors only give credit to Das Sarma's group [26-29] which does not properly reflect contributions from other groups. These ideas date back to 2012 (Kells et al Phys. Rev. B 86, 100503(R), 2012 and Prada et al, 86, 180503(R), 2012). The reviews cited in the introduction are bit outdated concerning state-of-the art. Here, I would recommend to cite R. Lutchyn et al, Nature Review Materials, 3, 52, 2018; R. Aguado, La Rivista del Nuovo Cimento, 40, 523, 2017). The papers I mentioned at the beginning of the report (Vuik et al, arXiv:1806.02801 and Avila et al, arXiv:1807.04677) should also be cited.

Corrected

In conclusion, this paper contains an interesting experimental setup that contains both edge and bulk tunneling probes in proximitized nanowires. This, in principle, allows to measure both the edge and bulk tunneling density of states (i. e Majoranas and gap closing). However, there are still important issues that need to be clarified in order to assess the importance and degree of novelty of these experiments, beyond incremental improvements, that would justify the claims in the paper and its publication in Nature Communications.

Our main purpose was to show that such bulk-edge correlation happens only under special conditions of: small diameter wires with a likely single-mode, and particular an ability to tune the chemical potential, without the deleterious effects of affecting the density at the bare edge of the wire (just outside the superconductor). Hence, we believe that we showed that only under special conditions the ZBCP is related to a Majorana state – but not always.

Referee 2

The authors are presenting tunnelling measurements of the tunnelling current into both the edge and the bulk of a InAs nanowire proximitized by aluminum coating and in the presence of a magnetic field. Under these conditions such a wire has been predicted to become topological and support Majorana states at these ends. These states should appear as zero-bias peaks in the tunnelling conductance at the ends of the wire, for a magnetic field larger than a critical value. Their appearance should correlate with the closing of the bulk gap and reopening as a topological gap.

Some of the experimental observations reported here are consistent with this description, in that the bulk is closing and reopening at values of the magnetic field consistent with those for which the zero-bias peak (ZBCP) appears in the end-tunnelling conductance. However, there many instances in which the values for which the bulk gap is closing are different than those at which the ZBCP appears (a possible explanation being a variation of the chemical potential from the 'sweet spot'), and more troubling, for some parameters ZPCPs appear both in the bulk and end spectra, indicating that the origin of the zero-energy states observed is non-topological.

The observed results are very important, but I believe that the main message of this work should be that an observed edge ZBCP is clearly not a direct indication for a Majorana state and more investigations are necessary. This casts even more doubt about the origin of many previous observations of ZBCPs and their interpretation as Majorana. The reverse argument, i.e. the observation of a ZPCP correlating with a closing and reopening of the bulk gap implies that such peak correspond to a Majorana state, is however in my opinion consistent with the data but not fully supported by it as the only possible explanation, since it seems that one needs quite a lot of fine tuning to find the right parameters to make it work and there is a lot of variation in the results. Since everything depends so much on the chemical potential, it is possible that one can find better alternative models and explanations that can describe well all the data, and that the correlation between the closing/opening of the gap with the formation of the ZBCP may as well be a coincidence with a mundane explanation, or it may well be the real thing.

However, I believe that the results presented here are very important and deserve publication since they show clearly that an end ZBCP may not necessarily be associated with a topological phase. Thus I would recommend publication provided that the message of the paper is made stronger in this direction.

We thank the Referee for acknowledging the importance of this work. We softened our main conclusions in the Abstract and summary of the paper, and thus hope it reflects better the obtained experimental data.

Referee 3

You find below your full report with our detailed comments attached (in *italic blue*).

We wish, right from the start, to commend the heroic effort and the time spent by the Referee to help us with improving the paper. This is quite an extraordinary effort on his part.

We are afraid though to say that a large fraction of the Referee's requests are not going to be fulfilled; a small number of them we disagree with while others (mostly data) we simply do not have. It is worth noting that - as many experimentalists recognize - working with wires is not easy; also reflected in this work.

I read the manuscript 'Concomitant opening of a topological bulk-gap with an emerging Majorana edge-state' by Grivnin et al. with great interest and I congratulate the authors with this achievement, as I am well aware this research is extremely challenging to successfully perform. The principal investigators in this work are leading a research team which has been part of the 'initial wave' of Majorana experiments in semiconducting nanowires in the 2012 period, and to the best of my knowledge, this presents their first subsequent paper on this topic. Experimentally speaking, this research field is still very narrow, and as such, independent experimental work as reported here, is crucial for the field's long term viability.

To the best of my knowledge, this is the first experimental report on tunnelling spectroscopy performed both at the end and in the middle of a possible topologically non-trivial phase in a semiconducting nanowire. This makes the work highly original. Semiconducting nanowires are in my opinion the leading candidate for demonstrating non-abelian statistics in the future, and I am therefore convinced this research is very important. Lastly, although I have significant criticism on the completeness and presentation of the data, the experiment and its outcome is of very high quality. Those three aspects are my motivation why I would firmly support the publication of a future improved version of this work in Nature Communications.

It is heartwarming and encouraging to read to opening statement.

Having said this, I find the current presentation of the work not publication ready, and surely not meeting the standards of Nature Communications. I therefore request significant improvements and modifications of the work. My main concern is the author's approach in their data selection to convincingly support the main conclusion of the work as expressed in their ambitious title. Clearly, they present (an) instance(s) in which the sought after behavior is observed. I therefore sympathize with the author's desire and attempt to make a grand claim with fitting title in this manuscript, this is how a research may get the right level of exposure.

However, naturally, this also means the research will be measured against an extremely ambitious standard fitting such a claim. I expect this main claim to be unequivocally demonstrated in a coherent fashion throughout the work. Instead, the whole body of data under consideration here, spread over different devices, cooldowns, and gate voltage configurations, presents itself to me as a rich, kaleidoscopic picture full of nuances, rather than the simple picture imposed on it by the work's title, which covers just one aspect of the observations.

As I outline below, it is my strong suspicion this rich behavior is caused by some form of localisation and the formation of Andreev bound states underneath each tunnel probe, causing complicated behavior among the 3 tunnel probes. As this type of experiment has been suggested to experimentalist by theoreticians for years, and as such has long been awaited within the community, I do not find this complexity an objection against the work's scientific merit, in fact, I am impressed and intrigued by this richness in the data. However, rather than oversimplifying this for the shallow reader by placing it under the title's umbrella, I think the Majorana research community would in the long turn benefit much more from a more complete and direct presentation and discussion of the fact that the sought after behaviour of an apparent zero bias conductance peak (zbc) at the wire's end 'concomitant' with the closing and reopening of a bulk gap is only one particular kind of behaviour found in this experiment, and it is very sensitive to gate voltage.

Indeed, our main purpose is to show that such bulk-edge correlation happens only under special conditions: small diameter wires with likely a single-mode and certain values of a chemical potential and magnetic field. Hence, we show that under other conditions this ideal behavior does not take place. We added a bulk-TP3 – placed further away – to show that disorder 'kills' the bulk-edge coincidence. We are showing the 'good' and 'bad' data (not only 'typical results') – and added more in the revised paper (in the Extended Data) – to stress that this experiment is not easy. Needless to say that we realized that this 'nakedness' would make the acceptance of the paper 'bumpier'.

I am highly appreciative of the fact that I am only able to have this opinion because the authors show such a wide range of behaviours in the combined main text and extended data. **However, I think that the authors should be less conservative in their approach, and address to some extent what I believe to be the elephant in the room** – it is unfortunately not 100% clear at this stage if a topologically non-trivial phase with Majorana zero modes at its ends has ever been experimentally demonstrated, and this work doesn't change that. This is not to say that no progress has been made, in fact, the community as a whole has by now gained an impressive level of understanding of the device physics, as demonstrated again in the quality of the results under consideration here. To me, this double message is the main lesson of the past 6 years of research inspired by the initial wave of experiments, **but it tends to get lost in triumphant one-liners such as this manuscripts title.**

*The statements marked in **bold** above are confusing. Are we **too bold** with a 'triumphant ... title' or we should be **more 'conservative'**?*

Provided the work's presentation improves significantly, I am convinced it deserves eventual publication in Nature Communication. The main aspects for the authors to focus on are:

1. a significant extension of the data included in the work, in particular, some basic device characterization data should be included in the extended data, the complementary gate dependencies to main text Figs 2 and 3 should be included in the main text, and whenever data on the main text device is shown, TP3's data should be included in cases where it has been omitted;
2. a critical revision of all references, I find there are important ones missing, and the function of others is unclear;
3. a thorough stylistic revision of text and figures.

In an attempt to aid the authors in this task, I have gone through their manuscript with great scrutiny, resulting in a long list of detailed comments and suggestions below. I expect the authors to respond to these in a satisfactory manner, and I genuinely hope this will result in a much improved, more balanced manuscript, which would benefit not only the authors but also the Majorana community as a whole, as I deem this work of significant importance to this research field.

In the following we will address the points raised by the Referee, leaving for him the judgement if they are satisfactory.

Detailed comments (I will use the line number of the pdf document in my comments):

Title:

For your consideration and left to your discretion, although the word 'concomitant' is very nice and well chosen, please be aware that its interpretation may not be entirely obvious to a broad international readership largely consisting of non-native English speakers.

More generally, I am wondering at this stage what would be a 'decisive' dataset given the author's experimental set-up that would legitimate this title. I am of the opinion that the conclusion of the presence of 'a topological bulk-gap' demands a level of robustness of these observations (i.e. 'Concomitant opening of a topological bulk-gap with an emerging Majorana edge-state') in parameter space. The authors demonstrate this as a function of magnetic field strength, but I don't think they find a very convincing degree of robustness of this phenomenon in gate voltage as the closest experimental proxy to chemical potential. If I am wrong, please show relevant additional data and I would strongly recommend putting this in the main text. If I am right, we can obviously think of many reasons why the sought after observations are very sensitive to gate voltage (I will raise one such possibility), and yet still correspond

to a topologically non-trivial phase, however, this is exactly the main claim of the title which I therefore expected to be demonstrated unequivocally without the need of all sorts of additional hypotheses. If the authors think I am wrong, please explain why. If the authors can think of another good experimental feature, beyond a wider extent in gate voltage range, that can demonstrate the main claim convincingly I'm happy to hear their reasoning and observe it in their data.

*We believe that the Title describes correctly our results. We do not claim of a **universal behavior**, but a coincidence between bulk and edge that requires delicate tuning – as generally expected in such nanowires. This becomes clear when the reader reads the (revised) Abstract. Maybe best for the Referee to go over the rest of our reply (and revisions), before addressing the issue of the Title again.*

Abstract

Overall I feel the abstract is appropriate, and I only have some minor comments.

5: I find this opening sentence a bit over the top. In theory, these zero-modes are surely immune to decoherence. In the experimental reality, it will be extremely difficult to protect these properly from nearby low energy states. Even this is achieved, still the problem of quasi-particle poisoning persists which will scramble up the computational basis provided by the majorana zero-modes. Although I surely think there is hope for engineering efforts to push this type of decoherence to long time scales, I find this sentence wrong and misleading, especially given this an experimental work.

We say 'are expected' – hence, theoretically. This is the motivation that drives the field (and Microsoft). If indeed this bothers the Referee, this can be replaced with: "...are expected to possess non-abelian statistics."

6: is the comma after the word superconductor correct?

15: 'for particular gate-tuning' – my main worry about the experimental evidence provided to back the title's claim. I find it too particular. See also later comments regarding this point, please leave this comment here as it is a crucial aspect of this work.

18-21: It may be necessary to slightly expand on this, given my later suggestions.

The Abstract was modified.

Introduction

24: I don't like the referencing here. I feel given this is an experimental work, the authors only need to cite the absolutely essential theoretical papers necessary to support this sentence. To me, these are ref 1, 2, 9, and 10. Although ref 3 more or less arrives at the same results as ref 1 and appeared around the same time, it completely lacks the vision expressed in ref 1 which turned out to be the inspiration

to the whole field, and it should therefore be omitted in my opinion. I feel the next crucial step was ref 2, where the insight was provided such systems can be engineered by combining other materials. Lastly, refs 9 and 10 give the actual 'recipe' implemented here. It may be useful to the interested reader to include some of reviews on this topic, such as ref 7, but it is not strictly necessary to backup this sentence.

25: Again I object to the referencing here. Clearly, the authors want to cite all experimental Majorana related works on semiconducting nanowires, which is to me indeed the right thing to do here. I find it particularly disturbing that the 2 papers by the De Franceschi research group are not cited here, nor anywhere else in the work. Those were very important experiments at the time, which made the whole community suddenly much more aware of the challenging nature of discriminating trivial zbcps from Majorana related zbcps. As this debate remains to some degree unresolved, they still serve that purpose and should be credited for that. Besides, some of the phenomena observed in the current manuscript may be explained by these mechanisms or closely related ones. Please include both works here, i.e. Lee et al. Nature Nanotechnology Vol 9 No 1 2014 and Lee et al. Phys. Rev. Lett. Vol 109 No 18 2012.

Furthermore, since all other experimental papers on Majorana fermions in nanowires are cited, surely the 2017 Science Advances paper by the Frolov research group should be included, Chen et al, Science Advances, Vol 3 No 9 2017.

32: the end of the sentence is in my opinion the appropriate place to cite ref 4.

32/33: please check language, I find the sentence somewhat inelegant. In terms of referencing, I suggest to not cite the two papers by Lee et al. at this point (please do include these in line 25), as they explicitly claim not to observe Majoranas. Please include the above mentioned paper by Chen et al. here as well.

36-38: I find using the phrase 'most crucial ingredients' arbitrary. They are indeed two essential ingredients, but I fail to see how these are more crucial than superconductivity if the whole idea falls apart with any of the 3 missing. Furthermore, although I agree with the spirit of the statement in the sense that independent verification of these aspect is lacking in the experiments, clearly attempts have been undertaken to show that these two ingredients are essential, namely by studying the zbcps behaviour while varying the magnetic field angle, or the gate potential.

39: I find the notion of 'nearly quantized' strange, I would not know how to define 'nearly quantized', nor how to discriminate it from 'not nearly quantized' or 'fully quantized'...

Revised

I am raising this because this notion gives the false impression that refs 26-29 where

highly original, which they aren't in my opinion - several important theory works predict trivial ZBCP's with a height of order G_0 and predate refs 26-29 by up to 5 years. Think of: Liu et al. Phys. Rev. Lett. 109.26 (2012): 267002; Bagrets and Altland Phys. Rev. Lett. 109, 227005, 2012; Pikulin et al. New Journal of Physics 14 2012; Mi et al. Journal of Experimental and Theoretical Physics 119, 6, 1018 2015. Surely some/all these references should be included here, and one could consider cutting down on refs 26-29.

Although refs 26-29 have some general relevance to this paper, a lot of the complicated behavior of ABS was covered earlier elsewhere already, in particular in works from Aguado and co-workers (e.g. ref 33, but there are some other relevant papers from those researchers), works from Loss and co-workers (e.g. ref 35), and even earlier works from Das Sarma and co-workers cover a lot of the physics reiterated in refs 26-29.

36-41: I find this passage as a whole somewhat weak. This research community would love to see a convincing experimental demonstration from a nanowire experiment of simultaneous tunnelling into the bulk and the end of a wire, confirming the Majorana hypothesis. Surely, a more compelling and pointy phrasing can be found to reflect this.

48: spelling 'Bohr magneton'

50: spelling.

Revised (with Refs. changes)

47: I got confused with the authors' definitions, please double check the following and point out the flaw in my reasoning or correct your own formulas and definitions: In the simplest model, the Majoranas originate from a pair of finite energy states, which (for $\mu = 0$) originate from the gap edges at $\pm\Delta$. Upon applying magnetic field, both states shift down with the Zeeman energy of a spin $\frac{1}{2}$ particle, i.e. $\pm \frac{1}{2} g \mu_B B$. That means to me the condition for the topological phase transition is reached when both states meet at the middle of the gap, i.e. when $\Delta = \frac{1}{2} g \mu_B B$, or for non-zero μ , when $\frac{1}{2} g \mu_B B = \sqrt{\Delta^2 + \mu^2}$. I would always define a Zeeman energy of a particle as the amount of energy it gains for a given magnetic field change. For spin $\frac{1}{2}$ particles, this is simply $\pm \frac{1}{2} g \mu_B B$. I find the current definition of E_z as twice that quantity confusing, an energy splitting is a difference between two energies and should incorporate some sort of delta symbol or so to stress that. Lastly, now both in line 47 and in line 53 an unnecessary factor 2 is incorporated.

*The difference in the factor of 2 is due to the definition of E_z . While the Referee define $E_z^{Referee} = \frac{1}{2} g \mu_B g B$ as the **Zeeman energy** we define it as the **Zeeman splitting** without the factor $\frac{1}{2}$ (see on page 2 of the manuscript "A sharp phase transition, which separates the topological and the trivial superconducting phases is expected to take*

place at $E_Z = 2\sqrt{\Delta_{ind}^2 + \mu^2}$ with Δ_{ind} the superconducting gap in the wire, $E_Z = g \mu_B B$, the Zeeman splitting, Landé g factor and μ_B the Bohr magneton. So that $E_Z^{Paper} = \mu_B g B = 2E_Z^{Referee}$, with the definition in the paper the experimental splitting between spin up and down states is equal to $E_Z^{Splitting} = E_Z^{Paper} = 2E_Z^{Referee} = 2E_Z^{Energy}$ both definitions are found in the literature and there is no contradictions.

Experimental set-up

Generally I find this section functional and having the right amount of detail. I have questions about the following:

72: Why refs 38-40? It seems to me these references were chosen without much care, or at least, they are poorly connected to the sentence after which they are placed. I presume the authors want to refer to some earlier work from the semiconducting nanowire research community to show how their work fits in that bigger picture? Ref 38 is on coherent electronic transport in VLS grown silicon nanowires. I see only very little relevance of ref 38 to this particular experiment. After all, the authors are not providing an extensive review of the experimental progress in semiconducting nanowire research, but are rather describing their own set-up. Ref 39 is irrelevant in my opinion, as it is a) about top down fabricated nanowires, not on bottom-up grown nanowires as considered here, and b) its not a III-V nanowire. Ref 40 is somewhat relevant, as it is at least on transport through III-V nanowires. However, the InP tunnel barriers in those devices have no relevance to this experiment. In any case, these 3 citations report experiments geared towards exploring rather different physics, i.e. creating single electron quantum dots. Or are the authors in a subtle way implying this is relevant to their set-up? I agree Ref 41 is relevant. Other relevant references on the proximity effect in group III-V nanowires can be found from the works in the late 2000's period in the Delft and Lund groups and it may be appropriate to cite these here.

73: It is not a very pressing aspect to clarify, but nevertheless, can the authors clarify why they are sure there is no conductance between TP's and Aluminium shell? I.e. do you know the typical tunnel resistance between tunnel probe and epitaxial Aluminium? Is this a self-terminating oxide formed by exposing the wire to air after growth? If yes, I tend to agree with the statement, as such an oxide can be used as a gate oxide separating two Aluminium gates, and is able to withstand up to a few volts of potential difference, typically. Consider making this more explicit.

It is well known that the native oxide on Al (some, 2-3nm thick) is an excellent barrier, thus preventing contacting. To contact the Al we had to etch away (or ion-mill) the oxide. The sentence was modified a bit.

76-77, and also 90-93 in the 'Results' section: I find these very bold claims that are not supported by any theoretical or experimental evidence whatsoever.

We do not see any boldness here. One of our intents in the fabrication of the devices was to prevent bare wire regions. We learnt over the years that the global back-gate, and even side-gates, affect the carrier density in the bare parts of the wire in comparison to the covered part by the superconductor.

My main concern and criticism towards this work is closely related to these statements. I would argue the opposite: the tunnel probes are effectively at 0 voltage. InAs nanowires tend to have their Fermi level pinned to the conduction band edge at the nanowire surface, i.e. the bandgap is shifted towards negative gate potentials. I presume this is why a negative backgate voltage is applied in the experiments, to go to a more interesting regime of somewhat low density in the nanowire. I would think that given their (almost) 0 V potential the tunnel probes would tend to attract charge carriers. The backgate potential may result in the generation of weak potential barriers in between the tunnel probes. This may very well lead to the formation of (weakly) localised Andreev bound states, bound by the superconducting Aluminium gap and the backgate induced weak barriers. As such, my first intuition would be to think of this experiment as 3 parallel 'Andreev quantum dots' which share the same common superconducting lead. I expect such an effect to vastly surpass the influence of any potential fluctuations due to defects within the wire or impurities on its surface. Obviously, my concern is of great impact on the interpretations of the results reported here, as in case of localized Andreev bound states underneath each TP, it is not very surprising that rich behavior is found upon comparing measurements from different TP's. It is not at all clear in such a scenario that a single, undisturbed topologically non-trivial phase is present in the device.

Can the authors please provide compelling experimental evidence why the picture I'm sketching is incorrect? If they can't, can they please remove such claims, and take my suggested picture (or some related variety), into careful consideration and rewrite the article from this perspective?

Many of the requests for additional data that will follow are closely related to this aspect.

Indeed this issue occupied us in the design and the fabrication of the devices. We thrived to have TPs with relative thick barriers, with high resistance and low capacitance TPs. This would have minimized the field penetration into the wire, and thus won't vary the carrier density. Devices with $\sim 100\text{k}\Omega$ resistance were found to be adequate (after many tries) – higher resistance devices lowered our sensitivity and thus also the resolution of a small signal. With the in situ grown Al SC, covering half the perimeter of the wire along its full length, flooding the wire with carriers and screening local potential fluctuations, the unbiased TPs, with Al oxide barrier, are not expected to alter in any significant way the potential underneath.

In a recent arXiv:1809.08250 by Das Sarma's group, the main subject of discussion was a comparison between invasive tunnel probes vs non-invasive ones.

We believe that our measurements were taken by ‘non-invasive’ probes. The induced potential under the tunnel probes, is likely to be weak and soft, thus may change the flow of the Andreev states in the wire, but do not lead to a localized potential well.

Moreover, the Referee would agree that if the TPs would have been be highly invasive, biasing negatively would deplete carriers, while biasing positively would accumulate them – a potential well will turn into a potential hill ! Yet, the spectroscopy we show is symmetric in both polarities of the TPs. Indeed we see a symmetric bulk-gap and the Andreev states – suggesting minimal invasiveness of the TPs.

And, after all, with a QD associated with a TP (as the Referee suggests), it is highly unlikely to observe the correlation we found.

*Is our argument provides a ‘**compelling experimental evidence**’?– We believe that it is a reasonable argument.*

79-82: This sentence implies that simultaneous measurements were performed on several tunnel barriers, and that this is depicted in Fig 1a. I see only a single voltage bias source in that picture. More generally, is this how you did your experiments? Or did you first perform an experiment on the edge TP with bulk TP left open, and then repeat it with the bulk TP leaving at TP open (or vice versa)? If you did a true simultaneous measurement of several TP’s, shouldn’t you place current meters in all branches of the circuit, i.e. on the input side of each TP, and as a sanity check on the (common) drain side as well? Either way, this statement or Fig 1a is incorrect. I think it is important that you make crystal clear everywhere in the manuscript which TP’s where biased and measured, and which ones where presumably left floating.

We did most of the measurements simultaneously, meaning sourcing from two TPs, edge and bulk, while placing a single current amplifier at the SC contact. This measurement configuration is possible when choosing two different frequencies for the two simultaneous measurements, so that they will not interfere (for example, 17Hz and 37Hz). The signal from the current amplifier is split and introduced to two lock-in amplifiers, each reading the current of its corresponding frequency (and thus TP). All other contacts of the device are left floating.

Obviously, before using this measurement configuration, we made sanity checks to be sure that having one or two voltage sources connected to the device do not affect the measured current of the corresponding TP.

Figure 1 was revised, making the experimental configuration clearer.

83: spelling, critical magnetic field I presume.

Corrected

Figure 1 + caption

Fig 1a - see comments above on measurement circuitry. I suspect your nanowires are hexagonal in cross sectional shape? If yes it would be better to depict it like that, a small effort to make the figure much more realistic.

Wires are round – not hexagonal (mentioned specifically in the Text).

Fig 1b - label TP's, contact, nanowire and include materials.

Fig 1c - I find this figure very unclear. There are many layers, many colors and most of them are left unlabelled. Again, if your wire has a hexagonal cross section please depict this. And are these large voids realistic? Please improve.

Fig 1d - Consider placing $B = 0$ T in the panel to make this very clear. 287-288: fix language, incomplete sentence.

295: Can you plot this on a log scale and add this to the extended data, to make clear there is indeed 2 or 3 of magnitude suppression between $V_{\text{bias}} = 0$ mV and $V_{\text{bias}} = \pm 0.3$ mV? If this is what you find, I would agree with this statement. Otherwise, data from a high resistive tunnel probe would be good to show in the extended data (e.g. a line trace both in linear and log scale at $B = 0$ T taken on the device on which extended data Fig 7 is taken).

Figure 1 was modified. Log-scale of Fig. 1d and of that of Fig. 7 in Extended Data (Fig. 10 in the revised version), were added to the Extended Data - now Fig. 11.

Results

I already expressed my general concerns regarding the interpretation of these data above. Here my detailed comments.

90-92: I do not understand this claim. In fact, I would think that the more opaque tunnel probe of the device shown in extended data Fig 7 was better, as its sharper nature clearly resulted in better resolution of the rich subgap features present. Also, I believe the tunneling conductance through a possible Majorana zero mode should scale with tunneling probability T , whereas for general Andreev bound states this rather scales with T^2 , i.e. measuring using more opaque barriers could be beneficial in suppressing tunneling into trivial states (obviously, it will suppress the zero bias peak height). Lastly, I find the comment ‘...while minimizing field penetration in the wire.’ rather cryptical. Please explain.

92-93: see above.

*We aimed at TPs' resistance of around 100kOhm. Yet, every added/subtracted monolayer of the Al oxide changed the TP's resistance dramatically. The higher the resistance and the lower the capacitance of the TP, more of the applied TP's voltage falls on the oxide and not in the wire. Please see our reply to the Referee comment in **bold** (above).*

96: language, some sort of verb seems missing ‘...to form a single ZBCP...’ sounds better to me

OK

101-102: No experiment can determine an exact gap closure / opening, it is

always limited by the measurement resolution (also in B field stepsize, by the way). I suggest instead of all the tildes in the preceding sentences, write $B = 0.7 \pm ? \text{ T}$ etc., and when you give a gap size at a certain field just write $E_g = 65 \pm ? \text{ microeV}$ at $B = 1.4\text{T}$. Seems more exact to me than the tildes.

102-104: I do not really get this comment and its relation to ref 45. Please clarify.

Revised due to these comments.

106-108: Agreed. I would go much further and argue that it is not obvious that what is measured at TP1, TP2 and TP3 is necessarily strongly correlated (beyond the obvious correlations given by sharing the same bulk gap), for above mentioned reasons of TP induced potential fluctuations.

The correlation we do find between TP1 and TP2, and (sometimes) the lack of it in TP3, suggests correlation length on the scale of TPs' distance (around 0.5 micron).

108-109: For your consideration: this is in my opinion not a very important point, as one can think of many reasons why the ZBCP may fluctuate in height as a function of global system parameters such as magnetic field. Also, although Extended Data Fig 1a appears convincing, it is somewhat deceptive, because around 1.4 T, already the bulk superconducting gap starts to be suppressed significantly. I think it is difficult to conclusively discriminate in a quantitative manner the increase in height of the ZBCP from an overall increase in conductance within the superconducting gap.

We present the actual data, which seems to agree with the expectation (even if the SC gap gets softer).

Figure 2 + caption

Consider placing something like 'Edge', 'Bulk - 1', 'Bulk - 2' above panels a, b, c. The little insets with device schematics are useful, but from just looking at the figures and schematic it may not be fully obvious that 'yellow' means this is the TP used in that scan. In panel c I find the dotted guides to the eye and vertical arrows obscuring the gap closure and reopening. Lastly, I strongly suggest to make plots similar to those shown in Extended data Fig 3 c and d (taking my comments on those into account) and place these in the extended data.

301: make clear what happens to the other two TP's when the other one is biased, i.e. presumably they are left floating?

303: its marked by short tick lines, not arrows.

Figure was modified. Waterfall plots of Fig. 2 were added to the Fig. 12 in the Extended Data.

Extended Data Figure 1 + caption

See my comment on line 108-109. I see no harm in leaving this Figure in the Extended

Data, but find its value somewhat limited. Also, the division in a green trivial and blue topological section depends heavily on the interpretation of this data. The authors could consider choosing a more neutral phrasing (although I agree some sort of partitioning of the behavior as a function of B is useful).

Please explicitly mention this is (presumably) based on the exact same dataset as plot in Fig. 2a and b, i.e. explicitly state panel a is a linecut at zero bias of Fig 2a and panel b is calculated based on the data of Fig 2b.

Done

Results - continued

110: I kindly request the authors to show additional data. I find the amount of data presented in this part of the study is insufficient to support many of the conclusions made in this work. I would like to see the following data:

1. Bias dependent tunneling spectroscopy vs Vgate for all 3 TP's, at a range of B fields starting from 0 T, with at least a few magnetic fields before the supposed topological phase transition, and a sufficient density of data right before, during and after this possible phase transition. Such data should focus on the low energy behavior, i.e. the subgap states, so a bias range of $\sim \pm 0.5$ mV clearly suffices. The gate range should surely include the whole range covered in Fig 2, 3 and Extended data Fig. 5 and 6. If such data is not available, I am somewhat sceptical about the overall scientific merit of the work, and my final recommendation for acceptance or rejection in Nature Communications will largely depend on the completeness of these data. Please anticipate a request from my side for inclusion of such data in main text and/or extended data.

2. Bias dependent tunneling spectroscopy vs Vgate for all 3 TP's with a specific focus on ruling out / confirming localisation effects. I find this manuscript lacks a set of basic characterisation data of the device's behavior as a function of gate. I think this should be part of the Extended Data, and surely shared with me given my concerns. I am thinking of a large scale conductance map at B = 0 T, ranging in gate space from pinch off (can you actually pinch off your devices? If not, I expect repetitive or stable behavior below some negative gate voltage?), until saturation at high gate voltage, with a sufficient bias voltage range, i.e. order 5 mV. I would like to see such a dataset for all 3 TP's.

3. Bias dependent tunneling spectroscopy vs magnetic field - the main text device has 3 TP's, and I would like to see the equivalent data of Fig 3 (now shown only for TP1 and TP2) also for TP3. Already Fig 2c shows that non-trivial difference between TP 2 and TP3 are present at fixed gate voltage, and I think Fig 3 should simply be expanded from 6 till 9 panels, including TP3 (unless the data is somewhat 'boring', in which case this should still be shown as Extended Data).

1. We found that scanning the back-gate - back and forth - causes charge instabilities, and a reliable measurement is achieved only by stepping the back-gate in the negative

direction (letting it rest) and scanning the magnetic field at each back-gate voltage step.

Please see below a sample of gate voltage dependence. The measurements were performed at $B=1.15T$. The left panel shows the conductance of the edge-TP1 as function of bias (x axis) and back-gate voltage (y axis). A robust ZBCP is present at a wide range of back-gate voltages.

The right panel shows the conductance of the bulk-TP2, measured with same parameters and at the same time as TP1. Here, we see a bulk gap is present at back-gate voltage range of $V_{BG} = -4.27V-4.19V$. We note that here the bulk gap is biggest when the ZBCP at the edge is strongest (here at $V_{BG} = -4.24V$, slightly shifted than the maximal gap obtained in the 'step-by-step' measurement presented in the main text, at $V_{BG} = -4.22V$). The ZBCP gets weaker with increased back-gate voltage and closure of the bulk gap. The measurement shows a few charge instabilities, a phenomenon common in most of experimentalists in the field.

2. We added a measurement vs. gate voltage at zero magnetic field to the Extended Data (Fig. 15); however, we do not have a large-scale conductance map, but rather scattered measurements we performed around zero bias.
3. A complimentary figure of TP3 dependence on the back-gate voltage was added in the Extended Data Fig. 14.

110-113: Unnecessary in my opinion, I am in agreement, but you provide no argument why this holds and the sentence isn't very exact. I don't think anybody will really doubt your backgate is having a very significant effect - any conductance line trace at fixed V_{bias} while varying V_{gate} will show huge variations, proving this point. Mind you that in my opinion you cannot rule out that tunnel strength may fluctuate as a function of V_{gate} - see comment on line 345-346 of Methods section below, however, this is a different point.

Since the tunneling barrier of the TPs are made of Al oxide, with a barrier height of some 3eV, we do not see how the back-gate voltage will affect this barrier. We wish to keep this comment in with a change in the wording.

116: 'Being the largest observed gap...' and 119: '...and never reopens in the full range of magnetic field (Fig. 3c).': typical statements that needs backing up by

additional data as requested under 1) above. You have the opportunity here to demonstrate the extend in V_{gate} of the ZBCP and reopened bulk gap, I strongly recommend doing so.

As stated above, the gap dependence on the back-gate voltage was measured and presented to Referee; however, not included into the paper due to instability issues.

122: Why do you have fluctuations in your effective g -factor? I am a bit sceptical about assigning effective g -factors to these states moving in B field. Clearly, they originate from a quasi-continuum at finite energy, where level repulsion causes a much ‘slower’ dispersion in B field. Often ‘along the way’ to zero energy other levels also disperse in B -field, which may lead to further distortions from a simple linear B field dependence. Lastly, how long a level ‘sticks’ to zero bias in magnetic field also heavily depends on repulsion from the nearby finite energy states. This influences assigning effective g -factors and points of ‘gap closure’ and ‘gap reopening’, and answering the naively binary question if a ZBCP is ‘robust’ or simply a cause of trivial ABS ‘sticking’ to zero bias for a while. Also, the coupling strength to the superconductor heavily influences all these behaviors. Ref 48 is quite illuminating in this respect and illustrates many of the points I am raising, you could consider citing this work in a more prominent spot in the text.

Fluctuations in the effective g -factor of sub-gap states occur when changing the electric field in the system (e.g. different back-gate voltage), as was shown in Vaitiekėnas et. al., 2018. Indeed, there might be a more complex B -dependence which we didn’t consider. Here we use a simple model, where we assume that the gap closes when the Zeeman energy is equal to the superconducting gap. The estimate of the g factor in this way should gives the correct order of magnitude. We add a sentence addressing that issue in the text.

122-123: Another statement that needs extensive back-up by providing gate dependent data as requested under 1) above.

As seen from the back-gate scans above, a ZBCP is present at the edge while either there is a reopened gap in the bulk, or a zero bias state present.

124-125: I appreciate the authors showing this behavior, as it constitutes a ‘negative’ result in the context of this paper’s main claim. Naturally, the question arises why any of the ZBCP’s behavior is strongly correlated with that of the ‘bulk gap’ measured at TP2. This connects to my overall comment of the possibility of probing individually localised states underneath each TP. A priori, this doesn’t necessarily mean all the reported behaviors of ZBCP and bulk gap are topologically trivial, but obviously a lot of nuancing seems very appropriated.

This issue had been addressed already above.

Figure 3 + caption

Please include data for TP3, unless it is identical to TP2 (Fig 2 doesn't suggest so), in which case it should still be included as part of the Extended Data in my opinion. I suggest to follow the same ordering of panels as in Fig 2, i.e. each row corresponds to the same gate voltage. That makes it much easier to compare behavior at different TP's for constant field. This would make Figure 3 a 9 panel figure. The authors could also consider combining Fig 2 and Fig 3 in a single Figure 2, and add a new Figure 3 with the complementary data of varying V_{gate} at constant B field. That would in my opinion be a much more comprehensive and consistent dataset.

A back-gate dependence of TP3 conductance vs. magnetic field was added to the Extended Data in Fig. 14.

As commented with Fig 2, consider placing something like 'Edge', 'Bulk - 1', 'Bulk - 2' above panels. I find several of the dotted 'guide-to-the-eye' lines obscuring nuances in the data, in particular in panel a around the crossing point of the states, for panel b around 1.3 T, and for panel f around 1.25T

A detail: In panel f, a switch is visible around 1.25T. Please share your opinion on the origin of this, and the occasional other switch visible throughout the data; is it an electron trap in the oxide (dis)charging, and if yes, it seems somewhat counterintuitive this happens as a function of magnetic field? How do you ensure such switches do not affect conclusions related to changing the chemical potential in the wire? Consider placing a short comment somewhere in the caption / main text/ methods section about the occasional occurrence of such switches (I congratulate the authors on achieving the level of stability displayed in the data throughout the paper, charge(?) switches seem rare which is great).

Lastly, I recommend to provide plots similar to those shown in Extended data Fig 3 c and d (taking my comments on those into account) and place these in the extended data.

The instabilities (can be of short and long time durations) are due to charging the dielectric of the tunnel probe (Al_2O_3) and global gate dielectric (SiO_2), or at their interfaces. We are certain that during the measurements presented in Fig. 3 there were no switches (except from the one seen in Fig. 3f), so that regardless of the effect of such a charge fluctuations, the conclusions of this part hold.

Needless to say that such 'jumps' can may shift the chemical potential (locally or globally) in the device; preventing a reliable gate-scanning measurements.

Plots were added in the Extended Data Fig. 13.

Results - continued

127-137: Although important from a viewpoint of completeness, I think this section and Figure 4 belong in the Extended Data. It is by now well established that the

temperature dependence of the ZBCP does not aid in uncovering its origin, and follows a trivial pattern of suppression as a function of T. Similarly, the ‘bulk gap’ closure appear rather trivial to me. I encourage the authors to focus instead on the in my opinion much more important and interesting questions of gate dependence and comparing all the different types of behavior observed within the main device, and across all devices.

It is here from the point of completeness.

Figure 4 + caption

See comment on lines 127-137 above. Panel a: There is still a tiny ZBCP left at 195 mK. Could you show a few more traces at higher temperatures to prove the ZBCP really disappears? Panel b: I find it somewhat strange you don’t show curves up to a similar T value as in panel a. Naturally, one now wonders if anything happens between the highest T value of 116 mK shown in panel b and the value of 195 mK shown in panel a. Lastly, it could be beneficial to duplicate the data as a colorplot, as different types of plotting draw attention to different types of features.

The highest temperature we measured with was 195mK. We believe that it is already clear that at this temperature the ZBCP nearly disappears. The features in bulk and edge were measured separately in this section (simultaneous measurement wasn’t performed since one lock-in was used for accurate temperature measurement and the other for conductance of the TP). The desired temperature was achieved via heating of the mixing chamber by a constant current and waiting for full equilibration, which was difficult to reproduce (accurately) in different measurements at different times. Lastly, the bulk-TP was not measured at higher temperatures since the small ‘shoulders’ already disappeared at 116mK.

The data in the color-plot presented below was interpolated for clarity (linear interpolation of the data presented in Fig. 4 with steps of 2mK). It doesn’t seem to add much information.

Summary

144-145: language, ZBCP is already defined, not sure what journal policy

recommends here. 'This is expected for the emergence...' seems better.

147-151: I feel this should be slightly expanded and rewritten based on the revisions of the paper I am suggesting.

Revised

Methods

Sample fabrication

This section is good as it is, except from one comment.

345-346: The effective tunnel barrier strength is not just dependent on the tunneling through the oxide (I indeed agree there's very little reason to expect such a tunnel strength will depend on the backgate voltage), but also depends on the exact probability density of the orbital nanowire wavefunction(s) near the tunnel barrier, which obviously will depend on the backgate voltage (although the effective 0V potential on the TP's may keep things fairly stable right underneath the TP's). In view of this I find this statement a bit misleading, it is anyway not necessary in this methods section.

Revised

Device parameters

354-355: If my earlier reasoning is correct, I would prefer defining E_z without the factor 2 in there.

See reply above

349-361: I find this estimate and the method used reasonable, and the lever arm found fits well with earlier experimental work on nanowire quantum dots (typically lever arm is of order 10% vs 1% in this work), especially considering the extra screening due the aluminum shell.

363: The authors find a change in charge of the order of 1 electron per micrometer of nanowire per 10 mV change in backgate. This again emphasises my earlier point of the possibility of localisation effects being very important in this work: your TP's have a width of ~ 0.2 microns, i.e. pretty much of the order of 1 micron. This compares quite favorably to the number you estimate (factor 5 smaller), given the many assumptions that goes into the estimate.

365-373: I think this is better left out. By now it is well established that classical capacitance calculations (miserably) fail in these systems and the appropriate approach is to self-consistently solve the Schrodinger-Poisson equation. See Vuik et al., New Journal of Physics, Volume 18, March 2016 and consider citing this paper, as it is relevant to this work.

It does not hurt to leave it in and stress again the disagreement with reality.

Extended data

Fig 1: see comments above.

The figure was revised.

Fig 2: Consider labelling panels with ‘edge’ and ‘bulk’. Can you extend the plot by another ~ 200 mT? It seems the gap hasn’t fully closed yet. Panel a: I find the ‘guide-to-the-eye lines somewhat obscuring the details. Given that the tunnel barrier is so transparent, are you even sure there is a ZBCP? Maybe it is just two levels coming very close and appearing as a broad ZBCP? Lastly, I recommend also plotting the linetraces for this data.

Below attached the revised plots, extended up to $B=1.5T$. Line-traces were added to the plot for clarity. At the edge, a ZBCP appears ($B=0.5T$), then splits and recombines (at $B=1.15T$). We think that the data may be interpreted by stabilization of the topological state at $B>0.7T$, as for that value there is a ZBCP at the edge and a gap in the bulk.

*However, the behavior at low field demands further interpretation, as it can result due to the high transparency of the edge probe (10 kOhm) and the short coherence length. However, following the Referee’s remark we agree that the evidence for topological state are not clear enough and therefore decide to remove the figure (see figure below) from the **Extended Data** in order to avoid confusion (as one can see we added a large number of figures to the Extended Data section).*

402-404: This is not a very strong statement, first you write that a ZBCP appears at 0.5 T, then that the bulk gap reopens at 0.6 T. I don't see a problem there? Having said that, I do indeed agree you may very well have had a different chemical potential for both tunnel probes.

Here we see the gap closing at $B=0.2T$. We expect that for a uniform chemical potential in both probes, a ZBCP should appear also at the edge at $B=0.2T$, and become stronger once the gap reopens. However, here the ZBP at the edge appears at higher field; hence, it is likely that the chemical potentials under the two TPs are different - but as noted above we decided to remove the figure, which may only confuse the reader.

More generally, is there more data on device 2, i.e. does it display other types of behavior for different gate voltages? It could be worth extending the figure with a few other types of behavior, if measured.

See reply above

Fig 3: Consider labelling panels a and b with 'edge' and 'bulk'. Panels c and d: I find the lines unclear because of all the individual dots. I am quite fine with just plotting a narrow line without individual datapoints. Also, I find the many colors not necessarily helpful, you could consider plotting it all in black and possibly indicate a few important traces by using a different color, such as onset of the ZBCP or point of gap closure / reopening.

Revised

417-421: Rather grand statement given the quality of the data. I fail to see: 1) why the ZBCP in panel a needs to be Majorana related, all I see is a gap closure pushing 2 pairs of ABS together to form a ZBCP right before superconductivity disappears. 2) a reopening of the gap in panel b, same arguments as for panel a). I find this a too large stretch. I would leave this dataset in the Extended Data, but give a more realistic down-to-earth description.

Revised

Same as for Fig 2 on device 2, is there more data that display other types of behavior for different gate voltages? It could be worth extending the figure with a few other types of behavior, if measured.

Yes, other types of behavior were measured. We have observed a ZBCP appearing at the edge with no bulk-gap closing and with a zero-bias state at the bulk at zero magnetic field. See added Fig. 6 and Fig. 9 in Extended Data.

Fig 4: Consider labelling panels a and b with 'edge' and 'bulk'.

Panel a): It seems a second pair of ABS 'merges' with the ZBCP at 0.9 T? Above that field value, the ZBCP is much broader, maybe there is a low energy 4 fold split conductance peak that gets washed out by the tunnel broadening?

panel b): Isn't there a weak ZBCP present at 1.2 T? Please comment.

The figure was revised. In both edge and bulk TPs there is an ABS that merges to a ZBCP (marked by dashed line in the positive bias side). Indeed, as the ABS merge at zero-bias, the ZBCP widens from $25\mu\text{eV}$ to $55\mu\text{eV}$, suggesting that the ABSs have an apparent finite energy due to level-repulsion smeared by tunnel-broadening and temperature.

Again, is there more data on device 1 after thermal cycling it, i.e. does it display other types of behavior for different gate voltages? Could you find a 'gap closure' and ZBCP appearance at similar fields as reported in Fig 2 and 3 main text? I recommend also plotting the linetraces for this data.

Yes, the thermally-recycled device also shows other types of behaviors, showing trivial ZBCPs at the edge with zero bulk-gap (added data in Extended Data Fig. 5 and Fig. 8, similar to the behavior in the Extended Data Fig. 4 and Fig. 7, respectively). TP3 was not measured in these cases.

Figs 2-4 form a useful additional dataset showing reproduction of the main features. In fact, I wouldn't mind extending the data in all 3 cases somewhat to cover a broader type of behaviors across the 3 devices.

Fig 5: I think this is an important dataset, that should be connected in some way to the results in the main text Figs 2 and 3. By changing the backgate voltage by a very small amount of 0.07V and 0.08V compared to the results in main text panels 3a,d and 3b,e respectively, suddenly this ZBCP at the edge became trivial. This really asks for some clarification. Is this the same feature as the one in the main text which is claimed to be non-trivial? If not, please proof this with experimental data, for example a continuous V_{gate} vs V_{bias} scan covering the whole range between main text Figs 2,3 and this figure, tracing the ZBCP. If it is the same feature, the authors should combine these data with those shown in the main text and make a single main text figure out of it. If it is not the same feature, I still strongly recommend to move this into the main text, as part of a separate figure that shows 'trivial' ZBCP's, with a corresponding discussion. Some more details: I think the data for TP3 should be included here as well (as with main text Fig 3). Label panels with 'edge', 'bulk 1', etc. Lastly typo in line 448, 'below' is what it should read.

The measurement in Fig. 5 (Fig. 4 in the revised version) shows a ZBCP appearing at the edge while the bulk gap closes and does not reopen (at least in our resolution) – at $V_{\text{BG}}=-4.29\text{V}$. This behavior is similar to the behavior presented at Figs. 3c & 3f in the main text with $V_{\text{BG}}=-4.13\text{V}$. It is trivial in the sense that there is no bulk-gap so the end-state is not topologically protected, exactly as in the case for $V_{\text{BG}}=-4.13\text{V}$ (Figs. 3c & 3f). Note, that on both sides of the 'sweet-spot', for higher and lower chemical potentials, the bulk-gap becomes smaller. Being similar to the data in Fig. 3 in the main text, it is better left in the Extended Data.

TP3 was not measured in this case.

Fig 6: Again, an important dataset. How does this connect (from an experimental viewpoint) to main text Figs 2 and 3, and the previous extended data Fig. 5? Do you have more data from neighboring gate values? If you can prove panel a) is a distinct feature from the one shown in main text Figs 2 and 3, it may still be worth showing these results in the main text as a separate figure. Isn't this another indication that what you measure at each TP isn't strongly related? Could you please include TP3's data as well?

The features that are shown here are different than what is shown in Figs. 2 & 3 – as the latter are in a different working point. Here, a ZBCP appears at the edge while the bulk shows a complicated magnetic field dependence of the ABS. We didn't measure continuously in the range $V_{BG}=-4.21V$ (Fig. 2) and $V_{BG}=-4.91V$ (Extended Data in Fig. 7). Here, we add, for the Referee, a measurement performed in the middle of this range, at $V_{BG}=-4.59V$ - showing a trivial behavior in the edge and in the bulk.

The edge shows a ZBCP throughout the full magnetic field range; hence, must be of trivial origin. TP3 was not measured in this data set.

Fig 7: I find this a very beautiful dataset, it should surely stay in the Extended data. Please consider my earlier remark, I don't really understand why you think this more opaque tunnel barrier is not suitable for edge tunneling spectroscopy. Please check the grammar in the caption there are several errors.

See an example of data taken with bulk-TPs with resistance in the Mega-ohm range. They barely show clear features below the SC gap. We could not sharpen or bring out better features.

Reviewers' comments:

Reviewer #1 (Remarks to the Author):

I am satisfied with the answer to all my comments except the second part of my second comment, which I copy now:

"More importantly, Fig. 3c clearly shows a robust ZBA after the gap closing, something unexpected in the bulk. Can the authors comment on this? "

The answer to this point is that owing to the finite energy resolution, of the order of $\sim 25\mu\text{eV}$, a very small gap may appear as a conductance peak. My question is then how do the authors distinguish trivial parity crossings from robust ZBAs? More specifically, what is the criterion that allows to interpret a peak like the one shown in Fig. 2a in terms of a robust ZBA and not some finite-energy crossing blurred by energy resolution? I think this important point deserves further discussion (both in the answer to my comment and in the manuscript).

Reviewer #3 (Remarks to the Author):

I thank the authors for their effort in improving the manuscript, as the new version constitutes a significant improvement in my opinion.

The main purpose of my initial response was to establish what is the most complete dataset the authors are able to present at this stage. I appreciate the author's effort on this end. As they point out themselves, they are only able to meet my requests for additional data to a limited extent, for various reasons. I was somewhat expecting this (but I had to ask) and I am sympathetic with the argument of the authors that at some point one must decide to accept a dataset as is and publish it, even if its imperfect, both for the sake of the people involved in the research but also as a message to the community that this is very challenging research. Judging the available data, I still stand by my initial opinion that this data is worthy of publication in Nature Communications, and I refer to my initial response for the arguments.

However, it must not come as a complete surprise to the authors that I find the data not convincing enough to support the main claim as stated in the title and at various points in the manuscript. I utterly fail to see why the observed occasional coincidence of appearance of a ZBCP at the edge and bulk gap closure must necessarily lead to the conclusion that these are topologically non-trivial effects. I would like to point out that this was the main criticism of all referees to this work in their initial response, and my opinion hasn't changed by this new version of the manuscript.

The results shown here nevertheless have a very clear value to the 'Majorana' community, mainly for two reasons. Firstly, it represents a high quality contribution, both in terms of experimental design and experimental outcome, independent of the main research consortium (Microsoft Station Q) active in the community, and this is important for the viability of the field. Secondly, as clearly stated by referee 2, this work is highly illuminating in demonstrating the great experimental difficulty in assigning 'Majoranas' to any experimentally observed ZBCP's, and 'Topological gaps' to any experimentally observed bulk gap. To use a term coined by referee 1, in my opinion, this clearly qualifies as a more than 'incremental progress', worthy of publication in Nature Communications.

My request to the authors is very simple and easy to implement: as I find their data at best suggestive, but not at all convincing, in demonstrating any topologically non-trivial effects, all such claims and suggestions should be removed from the manuscript (particularly in the title), except from the discussion section. In the discussion section, I expect them to write a short, balanced and nuanced piece where I'm happy for them to clearly state that the observed coincidence of appearing ZBCP and closing of the bulk gap is suggestive of topologically non-trivial effects, as long as the next statement explicitly mentions that it cannot be ruled out their observations are caused by disorder, possibly a TP induced disorder. If the authors are willing to do this, I will obviously recommend acceptance of this manuscript pending a quick final review. Although it would somewhat pain me to do so, in case the authors are not willing to do this, I see no other option but to recommend rejection.

In what follows I will focus on two aspects: some comments on the author's responses to my initial comments from the first review round to motivated my statements above, and I will explicitly point out the instances where I expect them to drop claims and remove suggestions concerning topological non-triviality.

Lastly, I would like to point out that the journal policy of Nature Communications is in strong support of a) publication of the raw experimental data, and b) publication of the referee reports along with the article. Regarding a), I'm a strong advocate of this myself, as I believe in complete transparency in this respect, and I urge the authors in case of acceptance to upload their data with the manuscript as a supplementary zip file, in some easily accessible format. Regarding b), again I am supportive, either with or without my identity being revealed. Ultimately, both aspects are to the editors and authors to decide upon, I am giving you my preference and recommendation.

Response to rebuttal: I will respond below in red.

List of requested changes: ***Please scroll down to the very end of my response!***

Editor

We suggest that this first paragraph should be also sent to all three Referees.

We thank the Referees for their comments and the time they invested in the reviews. We hope we've clarified most of the issues, and emphasized our main message – the importance of simultaneous bulk and edge measurement. We have been working with nanowires for some time and recognized (like many experimentalists do) the difficulties with working with them: their uncontrolled disorder, their 'switching' behavior, the non-reproducibility from wire to wire, and the difficult in controlling the chemical potential. The work we describe in this paper was a two-year effort of Ph.D. and M.Sc. students, who fabricated and tested many wires. The paper presents the best we can offer.

I understand and agree with the author's reasoning, and I appreciate the PI's spirit here.

As the paper may not be ideal, it may serve as beacon –being the first type of its kind - to similar (and maybe better) future works. This was also the case with the first nanowire/Majorana papers – being far from perfect, they ignited the candle that still burns.

Again, I largely agree, except for one aspect: in 2012, the community was generally naive and optimistic in the interpretation of the initial data (which was indeed the right thing to do as (1) all the nuances and difficulties known now weren't known back then and (2) it kicked off and accelerated the field tremendously). It is 2018 now, and we have all learned things are much more complicated, as this work also clearly demonstrates. As clearly stated by referee 2, the 'beacon' function of this work lies much more in exposing the great difficulty in finding solid evidence for Majoranas, a completely underexposed aspect in the experimental publication record of the field.

Referee 1

You find below your full report with our detailed comments attached (in *italic blue*).

First and foremost, the authors want to compare ZBAs measured with TP1 with bulk closing at TP2 and TP3. Surprisingly, they don't show clear cuts at $B=0$ to demonstrate that the spectral features of the sample (including of course the gap) are the same for TP1, TP2 and TP3. This is very relevant because they want to prove concomitance between ZBAs and gaps closing. The obvious question is therefore whether this "concomitance" does indeed correspond to the same physical situation. Such plots would also be important to identify the correct value of the gap in the proximitized wire (as compared to the Al gap). They write in the manuscript that the induced gap in the wire is close to the Al gap but this is not clearly seen in the contour plots (and clearly depends on TP, compare Fig 2a with Fig 2c at low B fields).

Line cuts at $B=0$ of TP1, TP2 and TP3 were added to Fig. 2 (in black), and also plotted in the figure below. Similar features appear in all three tunnel probes (marked by Δ , \square , \circ). The different tunnel probes show these features at slightly different energies and conductance values.

Note that the softness of the features and the overall conductance are very sensitive to the TP's resistance. The three TPs have slightly different resistances, which in turn result in a slightly different tunneling conductance values and appearance.

The two bulk-TPs have the same induced gap, while in the edge-TP it is slightly smaller. This difference results from the fact that TP1 sits at the edge of the wires – making the induced gap smaller - while the other are in the bulk. We see such a behavior in other devices too. For example, device #2 where the induced gap at the edge is $227 \pm 5 \mu\text{eV}$ and at the bulk it is $243 \pm 5 \mu\text{eV}$.

Related to the previous point: the authors themselves acknowledge that TP3 corresponds to a different chemical potential configuration which, I think, prevents from using Fig 2c (TP3) in comparison with Fig 2a (TP1). More importantly, Fig. 3c clearly shows a robust ZBA after the gap closing, something unexpected in the bulk. Can the authors comment on this?

The goal of Fig. 2 is to present the tunneling spectroscopy in different locations in the wire and thus point out the effect of unavoidable disorder. TP3 shows (in general) similar behavior to that of TP2 with a small shift in the magnetic field where the gap closes and reopens – such a shift reflects the slightly different chemical potential. Yet, the overall transition to a topological regime (everywhere) takes place for $B > 1.4\text{T}$.

In Fig. 3c we observe a gap closure without reopening as the chemical potential increases. We believe that in this configuration the chemical potential is higher than in Fig. 3a and 3b; hence, re-opening of a topological gap requires a Zeeman field that is near the critical

field. Note that our energy resolution is finite, being $\sim 25\mu\text{eV}$, and a very small gap may appear as a conductance peak.

Something similar occurs in Fig 3c where the bulk probe clearly shows a ZBA (what the authors call “absence of reopening”). This bulk measurement also shows traces of the ABSs present in fig 3f (almost parallel faint lines running at low bias). This seems to suggest that TP2 is not really a probe of the bulk only. Surprisingly, Fig 3f does not show a clear ZBA.

Figures 3c and 3f present a configuration in which a ZBCP appears at the edge of the sample but the bulk-gap is effectively zero (being smaller than our resolution). Figure 3f was indeed not clear enough because of a small jump that appears around $B=1.25T$. Line-cuts were added to the plot and show a clear ZBCP at the edge of the wire.

As noted correctly, both TP1 and TP2 show similar traces of ABS at low bias. We believe that these ABS exist throughout the entire wire, edge and bulk, thus seen in the tunnel probes. This emphasizes our main claim that observing “sticking of Andreev states” at the edge of the NW that leads to a robust ZBCP do not provide sufficient evidence for a Majorana ZBCP.

The authors support their claims with a second device (extended Fig. 2). They claim that the same concomitant behavior is seen here but I only see a parity crossing of ABSs (the “closing gap” in Fig. b is just a couple of sub gap states well inside a bulk gap at around $\pm 0.2\text{mV}$). Furthermore, no clear ZBA is seen for TP1.

*We thank the Referee for this comment. We have reexamining the data and add cuts for clarity, we think that the data may be interpreted by stabilization of the topological state at $B>0.7T$, as for that value there is a ZBCP at the edge and not in the bulk. However, the behavior at low field demand further interpretation, as it can result due to the high transparency of the edge-TP (10 kOhm) and the short coherence length. However, following the Referee’s remark we agree that the evidence for topological state are not clear enough and therefore decide to remove the figure (see figure below) from the **Extended Data** in order to avoid confusion (as one can see we added a large number of figures to the Extended Data section).*

The authors claim that their geometry avoids smooth potential/quantum dot effects. Yet, they show trivial ABS situations in the extended data (extended Fig.5) which are very similar to what one expects for such smooth/QD configurations. What is the origin of these ABSs? Can the authors comment on this?

Our devices are covered with in situ grown Al SC along half the perimeter of the wire, flooding the wire with carriers and screening local potential fluctuations (no bare wire is exposed). This is the reason we believe a QD configuration is less likely at the end of the wire due to a density depletion resulting from the global gate – as natural in customary fabricated devices. While less likely, we show an ABS appearing due to a local potential minimum due to disorder. This intends to show how important is to perform correlated measurements.

In the extended Fig. 7 they show a complex sub gap structure for bulk probes only. I am not sure if I understand the point that they try to make with such plots (apart from saying that all the spectral features and sub gap structure strongly depend on parameters).

The goal of Fig. 7 (Fig. 10 in the revised version of the Extended Data) is to show the potential spectroscopic abilities of such tunneling probes. Here, we wanted to show that

these tunnel probes allow in general a very clear and sharp measurement of the DOS in the nanowire.

Concerning samples, the wires studied here are extremely long (around six microns) which makes it very difficult that transport is ballistic across the whole wire. Moreover, the distance between tunneling probes is non-negligible (more than half a micron according to the SEM micrograph). How do the authors know that measurements at different tunneling probe electrodes correspond to the same (=ballistic) physical system? Is disorder over such long distances not an issue here?

The length of the entire wire is indeed long and we don't assume the transport is ballistic all along. Our goal was to measure transport at different positions on the wire and estimate the disorder length. Indeed, we show a much better correlation between edge-TP1 and the nearby bulk-TP2, and a shifted chemical potential at the further bulk-TP3. The overall length between TP1 and TP2 is $\sim 0.5\mu\text{m}$ - being similar to other 'Majorana type experiments' (e.g. Zhang et al. 2018). It's important to note that the nearest bulk-TP should be far enough from the edge so that there will be no overlap of the edge Majorana state (and collapsed gap) with that of the bulk's gap.

Some minor points: when referring to "trivial" ABS explanations the authors only give credit to Das Sarma's group [26-29] which does not properly reflect contributions from other groups. These ideas date back to 2012 (Kells et al Phys. Rev. B 86, 100503(R), 2012 and Prada et al, 86, 180503(R), 2012). The reviews cited in the introduction are bit outdated concerning state-of-the-art. Here, I would recommend to cite R. Lutchyn et al, Nature Review Materials, 3, 52, 2018; R. Aguado, La Rivista del Nuovo Cimento, 40, 523, 2017). The papers I mentioned at the beginning of the report (Vuik et al, arXiv:1806.02801 and Avila et al, arXiv:1807.04677) should also be cited.

Corrected

In conclusion, this paper contains an interesting experimental setup that contains both edge and bulk tunneling probes in proximitized nanowires. This, in principle, allows to measure both the edge and bulk tunneling density of states (i. e Majoranas and gap closing). However, there are still important issues that need to be clarified in order to assess the importance and degree of novelty of these experiments, beyond incremental improvements, that would justify the claims in the paper and its publication in Nature Communications.

Our main purpose was to show that such bulk-edge correlation happens only under special conditions of: small diameter wires with a likely single-mode, and particular an ability to

tune the chemical potential, without the deleterious effects of affecting the density at the bare edge of the wire (just outside the superconductor). Hence, we believe that we showed that only under special conditions the ZBCP is related to a Majorana state – but not always.

Referee 2

The authors are presenting tunnelling measurements of the tunnelling current into both the edge and the bulk of a InAs nanowire proximitized by aluminum coating and in the presence of a magnetic field. Under these conditions such a wire has been predicted to become topological and support Majorana states at these ends. These states should appear as zero-bias peaks in the tunnelling conductance at the ends of the wire, for a magnetic field larger than a critical value. Their appearance should correlate with the closing of the bulk gap and reopening as a topological gap.

Some of the experimental observations reported here are consistent with this description, in that the bulk is closing and reopening at values of the magnetic field consistent with those for which the zero-bias peak (ZBCP) appears in the end-tunnelling conductance. However, there are many instances in which the values for which the bulk gap is closing are different than those at which the ZBCP appears (a possible explanation being a variation of the chemical potential from the 'sweet spot'), and more troubling, for some parameters ZBCPs appear both in the bulk and end spectra, indicating that the origin of the zero-energy states observed is non-topological.

The observed results are very important, but I believe that the main message of this work should be that an observed edge ZBCP is clearly not a direct indication for a Majorana state and more investigations are necessary. This casts even more doubt about the origin of many previous observations of ZBCPs and their interpretation as Majorana. The reverse argument, i.e. the observation of a ZBCP correlating with a closing and reopening of the bulk gap implies that such peak correspond to a Majorana state, is however in my opinion consistent with the data but not fully supported by it as the only possible explanation, since it seems that one needs quite a lot of fine tuning to find the right parameters to make it work and there is a lot of variation in the results. Since everything depends so much on the chemical potential, it is possible that one can find better alternative models and explanations that can describe well all the data, and that the correlation between the closing/opening of the gap with the formation of the ZBCP may as well be a coincidence with a mundane explanation, or it may well be the real thing.

However, I believe that the results presented here are very important and deserve publication since they show clearly that an end ZBCP may not necessarily be associated with a topological phase. Thus I would recommend publication provided that the message of the paper is made stronger in this direction.

We thank the Referee for acknowledging the importance of this work. We softened our main conclusions in the Abstract and summary of the paper, and thus hope it reflects better the obtained experimental data.

Referee 3

You find below your full report with our detailed comments attached (in *italic blue*).

We wish, right from the start, to commend the heroic effort and the time spent by the Referee to help us with improving the paper. This is quite an extraordinary effort on his part.

My pleasure, it reflects my passion for the research field and I genuinely think the experiment and its outcome are valuable and worthy of publication in Nature Communications, regardless the in my opinion inconclusive nature of the results. Please consider my response to the TP's (non-)invasiveness carefully in particular.

We are afraid though to say that a large fraction of the Referee's requests are not going to be fulfilled; a small number of them we disagree with while others (mostly data) we simply do not have. It is worth noting that - as many experimentalists recognize - working with wires is not easy; also reflected in this work.

See above, I accept this and endorse the reasoning behind it.

I read the manuscript 'Concomitant opening of a topological bulk-gap with an emerging Majorana edge-state' by Grivnin et al. with great interest and I congratulate the authors with this achievement, as I am well aware this research is extremely challenging to successfully perform. The principal investigators in this work are leading a research team which has been part of the 'initial wave' of Majorana experiments in semiconducting nanowires in the 2012 period, and to the best of my knowledge, this presents their first subsequent paper on this topic. Experimentally speaking, this research field is still very narrow, and as such, independent experimental work as reported here, is crucial for the field's long term viability.

To the best of my knowledge, this is the first experimental report on tunnelling spectroscopy performed both at the end and in the middle of a possible topologically non-trivial phase in a semiconducting nanowire. This makes the work highly original. Semiconducting nanowires are in my opinion the leading candidate for demonstrating non-abelian statistics in the future, and I am therefore convinced this research is very important. Lastly, although I have significant criticism on the completeness and presentation of the data, the experiment and its outcome is of very high quality. Those three aspects are my motivation why I would firmly support the publication of a future improved version of this work in Nature Communications.

It is heartwarming and encouraging to read to opening statement.

Having said this, I find the current presentation of the work not publication ready, and surely not meeting the standards of Nature Communications. I therefore request significant improvements and modifications of the work. My main concern is the author's approach in their data selection to convincingly support the main conclusion of the work as expressed in their ambitious title. Clearly, they present (an) instance(s) in which the sought after behavior is observed. I therefore sympathize with the author's desire and attempt to make a grand claim with fitting title in this manuscript, this is how a research may get the right level of exposure. However, naturally, this also means the research will be measured against an extremely ambitious standard fitting such a claim. I expect this main claim to be unequivocally demonstrated in a coherent fashion throughout the work. Instead, the whole body of data under consideration here, spread over different devices, cooldowns, and gate voltage configurations, presents itself to me as a rich, kaleidoscopic picture full of nuances, rather than the simple picture imposed on it by the work's title, which covers just one aspect of the observations.

As I outline below, it is my strong suspicion this rich behavior is caused by some form of localisation and the formation of Andreev bound states underneath each tunnel probe, causing complicated behavior among the 3 tunnel probes. As this type of experiment has been suggested to experimentalist by theoreticians for years, and as such has long been awaited within the community, I do not find this complexity an objection against the work's scientific merit, in fact, I am impressed and intrigued by this richness in the data. However, rather than oversimplifying this for the shallow reader by placing it under the title's umbrella, I think the Majorana research community would in the long turn benefit much more from a more complete and direct presentation and discussion of the fact that the sought after behaviour of an apparent zero bias conductance peak (zbcp) at the wire's end 'concomitant' with the closing and reopening of a bulk gap is only one particular kind of behaviour found in this experiment, and it is very sensitive to gate voltage.

Indeed, our main purpose is to show that such bulk-edge correlation happens only under special conditions: small diameter wires with likely a single-mode and certain values of a chemical potential and magnetic field. Hence, we show that under other conditions this ideal behavior does not take place. We added a bulk-TP3 – placed further away – to show that disorder 'kills' the bulk-edge coincidence. We are showing the 'good' and 'bad' data (not only 'typical results') – and added more in the revised paper (in the Extended Data) – to stress that this experiment is not easy. Needless to say that we realized that this 'nakedness' would make the acceptance of the paper 'bumpier'.

I am highly appreciative of the fact that I am only able to have this opinion because the authors show such a wide range of behaviours in the combined main text and extended data. **However, I think that the authors should be less conservative in their approach, and address to some extent what I believe to be the elephant in the room** – it is unfortunately not 100% clear at this stage if a topologically non-trivial phase with Majorana zero modes at its ends has ever been experimentally demonstrated, and this

work doesn't change that. This is not to say that no progress has been made, in fact, the community as a whole has by now gained an impressive level of understanding of the device physics, as demonstrated again in the quality of the results under consideration here. To me, this double message is the main lesson of the past 6 years of research inspired by the initial wave of experiments, **but it tends to get lost in triumphant one-liners such as this manuscripts title.**

The statements marked in bold above are confusing. Are we too bold with a 'triumphant ... title' or we should be more 'conservative'?

I don't think my statements are that confusing. I want the authors to be less conservative in stating that their observations, although partially in line with a topologically non-trivial interpretation, are ultimately inconclusive, as no robustness of this behavior is observed, and clearly contradictory behavior has been observed at nearby chemical potential values. You were too conservative here in your representation, in my opinion.

I found and still find your title too bold given the inconclusive nature of the experiment.

So, to be clear: you are too bold and should be more conservative in respect to your conclusions around observations of topologically non-trivial physics. On the other hand, you should be much bolder and less conservative in stating the difficulties surrounding interpretation of your data, and exposing the experiment's inconclusive nature, and the relevance of this outcome to the vast majority of tunneling spectroscopy works out there.

Although not stated as directly in my initial comment above, I fail to see how my statement could have been interpreted any differently, but this clarification hopefully takes away any remaining confusion.

Provided the work's presentation improves significantly, I am convinced it deserves eventual publication in Nature Communication. The main aspects for the authors to focus on are:

1. a significant extension of the data included in the work, in particular, some basic device characterization data should be included in the extended data, the complementary gate dependencies to main text Figs 2 and 3 should be included in the main text, and whenever data on the main text device is shown, TP3's data should be included in cases where it has been omitted;
2. a critical revision of all references, I find there are important ones missing, and the function of others is unclear;
3. a thorough stylistic revision of text and figures.

In an attempt to aid the authors in this task, I have gone through their manuscript with great scrutiny, resulting in a long list of detailed comments and suggestions below. I expect the authors to respond to these in a satisfactory manner, and I genuinely hope this will result in a much improved, more balanced manuscript, which would benefit not only the

authors but also the Majorana community as a whole, as I deem this work of significant importance to this research field.

In the following we will address the points raised by the Referee, leaving for him the judgement if they are satisfactory.

Detailed comments (I will use the line number of the pdf document in my comments):

Title:

For your consideration and left to your discretion, although the word ‘concomitant’ is very nice and well chosen, please be aware that its interpretation may not be entirely obvious to a broad international readership largely consisting of non-native English speakers.

More generally, I am wondering at this stage what would be a ‘decisive’ dataset given the author’s experimental set-up that would legitimate this title. I am of the opinion that the conclusion of the presence of ‘a topological bulk-gap’ demands a level of robustness of these observations (i.e. ‘Concomitant opening of a topological bulk-gap with an emerging Majorana edge-state’) in parameter space. The authors demonstrate this as a function of magnetic field strength, but I don’t think they find a very convincing degree of robustness of this phenomenon in gate voltage as the closest experimental proxy to chemical potential. If I am wrong, please show relevant additional data and I would strongly recommend putting this in the main text. If I am right, we can obviously think of many reasons why the sought after observations are very sensitive to gate voltage (I will raise one such possibility), and yet still correspond to a topologically non-trivial phase, however, this is exactly the main claim of the title which I therefore expected to be demonstrated unequivocally without the need of all sorts of additional hypotheses. If the authors think I am wrong, please explain why. If the authors can think of another good experimental feature, beyond a wider extent in gate voltage range, that can demonstrate the main claim convincingly I’m happy to hear their reasoning and observe it in their data.

*We believe that the Title describes correctly our results. We do not claim of a **universal behavior**, but a coincidence between bulk and edge that requires delicate tuning – as generally expected in such nanowires. This becomes clear when the reader reads the (revised) Abstract. Maybe best for the Referee to go over the rest of our reply (and revisions), before addressing the issue of the Title again.*

It is my opinion that in tunneling spectroscopy experiments, the only way to legitimize a conclusion towards a topologically non-trivial nature of observed effects is a degree of robustness over other possible effects. This is obviously not the same as universal behavior, it just means some extent in experimental phase space of observed effects is required, surpassing those of trivially caused effects. Needless to say that this is very difficult to prove experimentally, and this is exactly why still controversy remains regarding interpreting such data as caused by topologically non-trivial physics. Your results are quite strikingly non-robust in gate voltage, in my opinion. This means that at best you are indeed probing a topologically

non-trivial phase, but it's a very delicate one, short lived in chemical potential and very sensitive to exact tuning. However, your experiment cannot discriminate this from a trivial explanation of some randomly coinciding peculiar effects of ABS crossing and sticking to zero bias (you show examples of states very robust in B field that must be trivial as they are already present at $B = 0$ T). Hence your title is not justifiable in my opinion, and you should change it to something more descriptive and less interpretative, boring as that may be. I give you a suggestion when I point out my requests for changes.

Abstract

Overall I feel the abstract is appropriate, and I only have some minor comments.

5: I find this opening sentence a bit over the top. In theory, these zero-modes are surely immune to decoherence. In the experimental reality, it will be extremely difficult to protect these properly from nearby low energy states. Even this is achieved, still the problem of quasi-particle poisoning persists which will scramble up the computational basis provided by the majorana zero-modes. Although I surely think there is hope for engineering efforts to push this type of decoherence to long time scales, I find this sentence wrong and misleading, especially given this an experimental work.

We say 'are expected' – hence, theoretically. This is the motivation that drives the field (and Microsoft). If indeed this bothers the Referee, this can be replaced with: "...are expected to possess non-abelian statistics."

Even theoretically decoherence due to quasi-particle poisoning is by now well-known and included into modelling I believe and I would argue that Majorana zero modes are therefore not expected to be free of decoherence. From a practical viewpoint (to which this opening statement tries to appeal, I guess) one decoherence mechanism (local interactions causing dephasing / relaxation of the qubit) or another (global interactions – quasi-particle poisoning randomizing the qubit basis) still have the same effect of rendering a future qubit useless. I agree that there is a strong interest and relevance to quantum computing, so I would suggest to just rephrase the statement slightly in that spirit avoiding the in my opinion misleading statement about insensitivity to decoherence. Something like 'Majorana zero modes are the prime candidate for realizing topological quantum computing' or so. On a different note, this current opening statement is pretty much unrelated to the rest of the abstract. Surely, a phrasing could be chosen that connects better to the actual experiment presented in this paper, yet still allows to touch upon the general relevance of the work?

Either way, it's a somewhat off topic discussion. I just raised it because it's the only sentence in the abstract I don't like, now and then. I otherwise feel the abstract is well balanced as is, although I have a minor suggestion at the end.

6: is the comma after the word superconductor correct?

15: 'for particular gate-tuning' – my main worry about the experimental evidence provided to back the title's claim. I find it too particular. See also later comments

regarding this point, please leave this comment here as it is a crucial aspect of this work.

18-21: It may be necessary to slightly expand on this, given my later suggestions.

The Abstract was modified.

Introduction

24: I don't like the referencing here. I feel given this is an experimental work, the authors only need to cite the absolutely essential theoretical papers necessary to support this sentence. To me, these are ref 1, 2, 9, and 10. Although ref 3 more or less arrives at the same results as ref 1 and appeared around the same time, it completely lacks the vision expressed in ref 1 which turned out to be the inspiration to the whole field, and it should therefore be omitted in my opinion. I feel the next crucial step was ref 2, where the insight was provided such systems can be engineered by combining other materials. Lastly, refs 9 and 10 give the actual 'recipe' implemented here. It may be useful to the interested reader to include some of reviews on this topic, such as ref 7, but it is not strictly necessary to backup this sentence.

25: Again I object to the referencing here. Clearly, the authors want to cite all experimental Majorana related works on semiconducting nanowires, which is to me indeed the right thing to do here. I find it particularly disturbing that the 2 papers by the De Franceschi research group are not cited here, nor anywhere else in the work. Those were very important experiments at the time, which made the whole community suddenly much more aware of the challenging nature of discriminating trivial zbcps from Majorana related zbcps. As this debate remains to some degree unresolved, they still serve that purpose and should be credited for that. Besides, some of the phenomena observed in the current manuscript may be explained by these mechanisms or closely related ones. Please include both works here, i.e. Lee et al. Nature Nanotechnology Vol 9 No 1 2014 and Lee et al. Phys. Rev. Lett. Vol 109 No 18 2012.

Furthermore, since all other experimental papers on Majorana fermions in nanowires are cited, surely the 2017 Science Advances paper by the Frolov research group should be included, Chen et al, Science Advances, Vol 3 No 9 2017.

32: the end of the sentence is in my opinion the appropriate place to cite ref 4.

32/33: please check language, I find the sentence somewhat inelegant. In terms of referencing, I suggest to not cite the two papers by Lee et al. at this point (please do include these in line 25), as they explicitly claim not to observe Majoranas. Please include the above mentioned paper by Chen et al. here as well.

36-38: I find using the phrase 'most crucial ingredients' arbitrary. They are indeed two essential ingredients, but I fail to see how these are more crucial than superconductivity if the whole idea falls apart with any of the 3 missing. Furthermore, although I agree with the

spirit of the statement in the sense that independent verification of these aspect is lacking in the experiments, clearly attempts have been undertaken to show that these two ingredients are essential, namely by studying the zbcps behaviour while varying the magnetic field angle, or the gate potential.

39: I find the notion of 'nearly quantized' strange, I would not know how to define 'nearly quantized', nor how to discriminate it from 'not nearly quantized' or 'fully quantized'...

Revised

Thank you for critically revising your references, I am satisfied by these improvements.

I am raising this because this notion gives the false impression that refs 26-29 were highly original, which they aren't in my opinion - several important theory works predict trivial ZBCP's with a height of order G_0 and predate refs 26-29 by up to 5 years. Think of: Liu et al. Phys. Rev. Lett. 109.26 (2012): 267002; Bagrets and Altland Phys. Rev. Lett. 109, 227005, 2012; Pikulin et al. New Journal of Physics 14 2012; Mi et al. Journal of Experimental and Theoretical Physics 119, 6, 1018 2015. Surely some/all these references should be included here, and one could consider cutting down on refs 26-29.

Although refs 26-29 have some general relevance to this paper, a lot of the complicated behavior of ABS was covered earlier elsewhere already, in particular in works from Aguado and co-workers (e.g. ref 33, but there are some other relevant papers from those researchers), works from Loss and co-workers (e.g. ref 35), and even earlier works from Das Sarma and co-workers cover a lot of the physics reiterated in refs 26-29.

36-41: I find this passage as a whole somewhat weak. This research community would love to see a convincing experimental demonstration from a nanowire experiment of simultaneous tunnelling into the bulk and the end of a wire, confirming the Majorana hypothesis. Surely, a more compelling and pointy phrasing can be found to reflect this.

48: spelling 'Bohr magneton'

50: spelling.

Revised (with Refs. changes)

47: I got confused with the authors' definitions, please double check the following and point out the flaw in my reasoning or correct your own formulas and definitions: In the simplest model, the Majoranas originate from a pair of finite energy states, which (for $\mu = 0$) originate from the gap edges at $\pm\Delta$. Upon applying magnetic field, both states shift down with the Zeeman energy of a spin $\frac{1}{2}$ particle, i.e. $\pm \frac{1}{2} g \mu_B B$. That means to me the condition for the topological phase transition is reached when both states meet at the middle of the gap, i.e. when $\Delta = \frac{1}{2} g \mu_B B$, or for non-zero μ , when $\frac{1}{2} g \mu_B B = \sqrt{\Delta^2 + \mu^2}$. I would always define a Zeeman energy of a particle as the amount of energy it gains for a given magnetic field change. For spin $\frac{1}{2}$ particles, this is simply $\pm \frac{1}{2} g \mu_B B$. I find the current definition of E_z as twice that quantity confusing, an energy splitting is a difference between two energies and should incorporate some sort

of delta symbol or so to stress that. Lastly, now both in line 47 and in line 53 an unnecessary factor 2 is incorporated.

*The difference in the factor of 2 is due to the definition of E_Z . While the Referee define $E_Z^{Referee} = 1/2 g \mu_B g B$ as the **Zeeman energy** we define it as the **Zeeman splitting** without the factor $1/2$ (see on page 2 of the manuscript "A sharp phase transition, which separates the topological and the trivial superconducting phases is expected to take place at $E_Z = 2\sqrt{\Delta_{ind}^2 + \mu^2}$ with Δ_{ind} the superconducting gap in the wire, $E_Z = g \mu_B B$, the Zeeman splitting, Landé g factor and μ_B the Bohr magneton. So that $E_Z^{Paper} = \mu_B g B = 2E_Z^{Referee}$, with the definition in the paper the experimental splitting between spin up and down states is equal to $E_Z^{Splitting} = E_Z^{Paper} = 2E_Z^{Referee} = 2E_Z^{Energy}$ both definitions are found in the literature and there is no contradictions.*

Ok. It's an unimportant detail, but I find it weird to label an energy splitting with the symbol 'E' which is quite universally reserved for an energy of an eigenstate, and not for an energy difference between two eigenstates. In your definition, I'd suggest to call it ' ΔE_Z ', to also symbolically emphasize you are looking at an energy difference.

Experimental set-up

Generally I find this section functional and having the right amount of detail. I have questions about the following:

72: Why refs 38-40? It seems to me these references were chosen without much care, or at least, they are poorly connected to the sentence after which they are placed. I presume the authors want to refer to some earlier work from the semiconducting nanowire research community to show how their work fits in that bigger picture? Ref 38 is on coherent electronic transport in VLS grown silicon nanowires. I see only very little relevance of ref 38 to this particular experiment. After all, the authors are not providing an extensive review of the experimental progress in semiconducting nanowire research, but are rather describing their own set-up. Ref 39 is irrelevant in my opinion, as it is a) about top down fabricated nanowires, not on bottom-up grown nanowires as considered here, and b) its not a III-V nanowire. Ref 40 is somewhat relevant, as it is at least on transport through III-V nanowires. However, the InP tunnel barriers in those devices have no relevance to this experiment. In any case, these 3 citations report experiments geared towards exploring rather different physics, i.e. creating single electron quantum dots. Or are the authors in a subtle way implying this is relevant to their set-up? I agree Ref 41 is relevant. Other relevant references on the proximity effect in group III-V nanowires can be found from the works in the late 2000's period in the Delft and Lund groups and it may be appropriate to cite these here.

73: It is not a very pressing aspect to clarify, but nevertheless, can the authors clarify why they are sure there is no conductance between TP's and Aluminium shell? I.e. do you

know the typical tunnel resistance between tunnel probe and epitaxial Aluminium? Is this a self-terminating oxide formed by exposing the wire to air after growth? If yes, I tend to agree with the statement, as such an oxide can be used as a gate oxide separating two Aluminium gates, and is able to withstand up to a few volts of potential difference, typically. Consider making this more explicit.

It is well known that the native oxide on Al (some, 2-3nm thick) is an excellent barrier, thus preventing contacting. To contact the Al we had to etch away (or ion-mill) the oxide. The sentence was modified a bit.

76-77, and also 90-93 in the 'Results' section: I find these very bold claims that are not supported by any theoretical or experimental evidence whatsoever.

We do not see any boldness here. One of our intents in the fabrication of the devices was to prevent bare wire regions. We learnt over the years that the global back-gate, and even side-gates, affect the carrier density in the bare parts of the wire in comparison to the covered part by the superconductor.

I agree with you that a half coverage of the wire is likely to largely mitigate the effect of a tunnel probe induced potential fluctuation. The question is if there is no effect whatsoever, which is what you claim, whereas I'm thinking how can you exclude that there may be just enough of an effect to create some additional localized ABS underneath or between tunnel probes.

My main concern and criticism towards this work is closely related to these statements. I would argue the opposite: the tunnel probes are effectively at 0 voltage. InAs nanowires tend to have their Fermi level pinned to the conduction band edge at the nanowire surface, i.e. the bandgap is shifted towards negative gate potentials. I presume this is why a negative backgate voltage is applied in the experiments, to go to a more interesting regime of somewhat low density in the nanowire. I would think that given their (almost) 0 V potential the tunnel probes would tend to attract charge carriers. The backgate potential may result in the generation of weak potential barriers in between the tunnel probes. This may very well lead to the formation of (weakly) localised Andreev bound states, bound by the superconducting Aluminium gap and the backgate induced weak barriers. As such, my first intuition would be to think of this experiment as 3 parallel 'Andreev quantum dots' which share the same common superconducting lead. I expect such an effect to vastly surpass the influence of any potential fluctuations due to defects within the wire or impurities on its surface. Obviously, my concern is of great impact on the interpretations of the results reported here, as in case of localized Andreev bound states underneath each TP, it is not very surprising that rich behavior is found upon comparing measurements from different TP's. It is not at all clear in such a scenario that a single, undisturbed topologically non-trivial phase is present in the device.

Can the authors please provide compelling experimental evidence why the picture I'm

sketching is incorrect? If they can't, can they please remove such claims, and take my suggested picture (or some related variety), into careful consideration and rewrite the article from this perspective?

Many of the requests for additional data that will follow are closely related to this aspect.

Indeed this issue occupied us in the design and the fabrication of the devices. We thrived to have TPs with relative thick barriers, with high resistance and low capacitance TPs. This would have minimized the field penetration into the wire, and thus won't vary the carrier density. Devices with $\sim 100\text{k}\Omega$ resistance were found to be adequate (after many tries) – higher resistance devices lowered our sensitivity and thus also the resolution of a small signal. With the in situ grown Al SC, covering half the perimeter of the wire along its full length, flooding the wire with carriers and screening local potential fluctuations, the unbiased TPs, with Al oxide barrier, are not expected to alter in any significant way the potential underneath.

See response above.

In a recent arXiv:1809.08250 by Das Sarma's group, the main subject of discussion was a comparison between invasive tunnel probes vs non-invasive ones. We believe that our measurements were taken by 'non-invasive' probes. The induced potential under the tunnel probes, is likely to be weak and soft, thus may change the flow of the Andreev states in the wire, but do not lead to a localized potential well.

I would have liked to get some evidence from your end that your 'believe' is based on actual experimental evidence. In the absence thereof, it's your word against mine, ultimately. I do think this is a serious aspects of consideration, and you could consider engaging with some of the theory groups to properly model your device in a self-consistent Schrodinger-Poisson solver taking into account your exact device geometry and potentials of backgate, tunnel probes and half-covering Al shell. E.g. recent results from the Delft theorists in collaboration with the experimentalist look quite promising for the intuition one gets from such a modelling exercise. That would allow you to comment somewhat more quantitatively on your tunnel probe's (non-)invasiveness. Such an effort is beyond the current manuscript's extent. I have reread your exact phrasing here, and I can accept your usage of the word 'mitigating' in line 81 of the revised manuscript as reflecting your believes in this respect, since that phrasing still leaves some room for the possibility of an invasive tunnel probe

Moreover, the Referee would agree that if the TPs would have been be highly invasive, biasing negatively would deplete carriers, while biasing positively would accumulate them – a potential well will turn into a potential hill ! Yet, the spectroscopy we show is symmetric in both polarities of the TPs. Indeed we see a symmetric bulk-gap and the Andreev states – suggesting minimal invasiveness of the TPs.

This argument is wrong in my opinion. Please try and follow my simplistic argument below:

The TP is at zero volt (when unbiased). The backgate is at minus several volts. I.e., the relevant scale of voltage difference here is of order 1V. I would argue that in a nanowire device, on the scale of the TP's width and separation, the relevant scale for localization effects to kick in is when potential fluctuations of the order of hundreds of microeV, i.e. order 1 meV, appear. For the TP, to cause a potential fluctuation of that order at a fixed backgate voltage and given the voltage difference of order 1 V, the required leverarm is of order 1 promille (obviously, we don't know what is the relevant exact 'voltage difference' here, but I think order 1V is the scale to be anticipated, intuitively). I am not surprised at all you find pretty symmetric (but not exactly symmetric!!!) tunneling spectroscopy data in the 1 mV biasing range, as such a small lever arm of 1 promille will only induce a shift in chemical potential of order 1 microeV, which I would argue is well below your experimental resolution. And indeed, for such a thin AlO₂ tunnel barrier, a 1 promille lever arm is very small, in line with your believe of the half shell mitigating potential fluctuations induced by the TP. But that doesn't mean at all your TP isn't inducing any potential fluctuation because of its voltage difference with respect to the backgate!

I am well aware this is a rather crude estimate I'm doing here, at the same time, it's pretty much the best one can do without resorting to my suggestion above of more advanced modelling.

Furthermore, mind you, I've been rather conservative in my estimates here, as there are several factors which 'mitigate' the effect of biasing the TP on the tunneling spectroscopy data. The SC gap continuum is always pinned to the Fermi level anyway, so for small shifts at a few hundred microV bias, I don't expect much variation for changing polarity if it comes to the SC gap continuum. So the only features that I would expect to depend on polarity of the bias are the actual features of interest, i.e. the subgap states. Slight asymmetry of such features is only well discernable deeper within in the gap, where rounding due to an onset of the gap edge continuum is absent. So likely only in the +- 100 microeV you'd be able to reliable discern such an asymmetry. Add to this the broadening caused by your lockin excitation, which sets your resolution to something like ~5 microeV, and you end up with a possible, non-discernible lever arm of the TP of 5%, which is very large compared to the types of leverarms I expect to be required for inducing a non-homogenous potential landscape!!!

Bottomline, you are clearly comparing apples and oranges here...

Lastly, using your own argument of observed symmetric tunneling spectroscopy traces, I actually do think that careful examination of the linetraces shown throughout the manuscript reveals that exact symmetry for low energy features is absent. There is in fact an observable difference in tunneling conductance at positive and negative bias, albeit small, mostly visible as a slight distortion of a sub-gap conductance peak upon comparing

positive and negative bias. This could very well indicate that my suspicion of your TP's having a non-negligible gating effect is correct!

And, after all, with a QD associated with a TP (as the Referee suggests), it is highly unlikely to observe the correlation we found.

For your reasoning to hold you assumed not only that non-invasive tunnel probes may lead to correlation, but also the reverse: observing correlation demands non-invasive tunnel probes. In the simplest continuum model I agree with this statement, provided the observed correlation is indeed of a topologically non-trivial nature. If the correlation is for trivial reasons, we can cook up many scenarios, including the one I've sketched, or just some lucky combination of careful tuning for a specific disorder configuration, etc etc. My point being, for the above statement to hold you already need to assume your correlation is non-trivial in nature. However, the whole point you are trying to prove in this paper is exactly that, hence your need to stress your tunnel probe is non-invasive, as a prerequisite to that conclusion. The above statement is therefore an unacceptable case of circular reasoning.

Is our argument provides a 'compelling experimental evidence'?– We believe that it is a reasonable argument.

I think its flawed, for reasons given above.

The point I really want to convey is I genuinely don't know if your tunnel probes have a non-negligible effect on what happens in the nanowire, or not, and I would love to know the answer. I am a bit concerned about it, obviously. Practically speaking, I think this shouldn't stop you from publishing the data as is in Nat Com, but you should pay attention to this in the article, I want you to bring this scenario up in the discussion, and point out further theory work is required to gain more insight here. I find your current approach too simplistic as you have not backed it up by any sort of evidence, and in fact, if there is any experimental evidence it may point in this direction (e.g. the slight asymmetries you observe in your conductance spectroscopy, and of course the general 'richness' of your data and its sensitivity to gate tuning). Once more, I don't see a principal problem with it, this is a very difficult aspect to assess and control experimentally. But just say it! That way, follow up research can look into it.

79-82: This sentence implies that simultaneous measurements were performed on several tunnel barriers, and that this is depicted in Fig 1a. I see only a single voltage bias source in that picture. More generally, is this how you did your experiments? Or did you first perform an experiment on the edge TP with bulk TP left open, and then repeat it with the bulk TP leaving at TP open (or vice versa)? If you did a true simultaneous measurement of several TP's, shouldn't you place current meters in all branches of the circuit, i.e. on the input side of each TP, and as a sanity check on the (common) drain side

as well? Either way, this statement or Fig 1a is incorrect. I think it is important that you make crystal clear everywhere in the manuscript which TP's were biased and measured, and which ones were presumably left floating.

We did most of the measurements simultaneously, meaning sourcing from two TPs, edge and bulk, while placing a single current amplifier at the SC contact. This measurement configuration is possible when choosing two different frequencies for the two simultaneous measurements, so that they will not interfere (for example, 17Hz and 37Hz). The signal from the current amplifier is split and introduced to two lock-in amplifiers, each reading the current of its corresponding frequency (and thus TP). All other contacts of the device are left floating.

Obviously, before using this measurement configuration, we made sanity checks to be sure that having one or two voltage sources connected to the device do not affect the measured current of the corresponding TP.

Figure 1 was revised, making the experimental configuration clearer.

Thanks for clarifying this.

83: spelling, critical magnetic field I presume.

Corrected

Figure 1 + caption

Fig 1a - see comments above on measurement circuitry. I suspect your nanowires are hexagonal in cross sectional shape? If yes it would be better to depict it like that, a small effort to make the figure much more realistic.

Wires are round – not hexagonal (mentioned specifically in the Text).

My bad.

Fig 1b - label TP's, contact, nanowire and include materials.

Fig 1c - I find this figure very unclear. There are many layers, many colors and most of them are left unlabelled. Again, if your wire has a hexagonal cross section please depict this. And are these large voids realistic? Please improve.

Fig 1d - Consider placing $B = 0$ T in the panel to make this very clear. 287-288: fix language, incomplete sentence.

295: Can you plot this on a log scale and add this to the extended data, to make clear there is indeed 2 or 3 of magnitude suppression between $V_{\text{bias}} = 0$ mV and $V_{\text{bias}} = \pm 0.3$ mV? If this is what you find, I would agree with this statement. Otherwise, data from a high resistive tunnel probe would be good to show in the extended data (e.g. a line trace both in linear and log scale at $B = 0$ T taken on the device on which extended data Fig 7 is taken).

Figure 1 was modified. Log-scale of Fig. 1d and of that of Fig. 7 in Extended Data (Fig. 10 in the revised version), were added to the Extended Data - now Fig. 11.

Results

I already expressed my general concerns regarding the interpretation of these data above. Here my detailed comments.

90-92: I do not understand this claim. In fact, I would think that the more opaque tunnel probe of the device shown in extended data Fig 7 was better, as its sharper nature clearly resulted in better resolution of the rich subgap features present. Also, I believe the tunneling conductance through a possible Majorana zero mode should scale with tunneling probability T , whereas for general Andreev bound states this rather scales with T^2 , i.e. measuring using more opaque barriers could be beneficial in suppressing tunneling into trivial states (obviously, it will suppress the zero bias peak height). Lastly, I find the comment ‘...while minimizing field penetration in the wire.’ rather cryptical. Please explain.

92-93: see above.

*We aimed at TPs’ resistance of around 100kOhm. Yet, every added/subtracted monolayer of the Al oxide changed the TP’s resistance dramatically. The higher the resistance and the lower the capacitance of the TP, more of the applied TP’s voltage falls on the oxide and not in the wire. Please see our reply to the Referee comment in **bold** (above).*

96: language, some sort of verb seems missing ‘...to form a single ZBCP...’ sounds better to me

OK

101-102: No experiment can determine an exact gap closure / opening, it is always limited by the measurement resolution (also in B field stepsize, by the way). I suggest instead of all the tildes in the preceding sentences, write $B = 0.7 \pm ? T$ etc., and when you give a gap size at a certain field just write $E_g = 65 \pm ? \text{ microeV}$ at $B = 1.4T$. Seems more exact to me than the tildes.

102-104: I do not really get this comment and its relation to ref 45. Please clarify.

Revised due to these comments.

106-108: Agreed. I would go much further and argue that it is not obvious that what is measured at TP1, TP2 and TP3 is necessary strongly correlated (beyond the obvious correlations given by sharing the same bulk gap), for above mentioned reasons of TP induced potential fluctuations.

The correlation we do find between TP1 and TP2, and (sometimes) the lack of it in TP3,

suggests correlation length on the scale of TPs' distance (around 0.5 micron).

As expected, both in the case of topologically non-trivial physics being the cause, and in case some disorder induced (either intrinsic, or via TP potentials) ABS happen to behave as observed upon careful tuning.

108-109: For your consideration: this is in my opinion not a very important point, as one can think of many reasons why the ZBCP may fluctuate in height as a function of global system parameters such as magnetic field. Also, although Extended Data Fig 1a appears convincing, it is somewhat deceptive, because around 1.4 T, already the bulk superconducting gap starts to be suppressed significantly. I think it is difficult to conclusively discriminate in a quantitative manner the increase in height of the ZBCP from an overall increase in conductance within the superconducting gap.

We present the actual data, which seems to agree with the expectation (even if the SC gap gets softer).

Figure 2 + caption

Consider placing something like 'Edge', 'Bulk - 1', 'Bulk - 2' above panels a, b, c. The little insets with device schematics are useful, but from just looking at the figures and schematic it may not be fully obvious that 'yellow' means this is the TP used in that scan. In panel c I find the dotted guides to the eye and vertical arrows obscuring the gap closure and reopening. Lastly, I strongly suggest to make plots similar to those shown in Extended data Fig 3 c and d (taking my comments on those into account) and place these in the extended data.

301: make clear what happens to the other two TP's when the other one is biased, i.e. presumably they are left floating?

303: its marked by short tick lines, not arrows.

Figure was modified. Waterfall plots of Fig. 2 were added to the Fig. 12 in the Extended Data.

Extended Data Figure 1 + caption

See my comment on line 108-109. I see no harm in leaving this Figure in the Extended Data, but find its value somewhat limited. Also, the division in a green trivial and blue topological section depends heavily on the interpretation of this data. The authors could consider choosing a more neutral phrasing (although I agree some sort of partitioning of the behavior as a function of B is useful).

Please explicitly mention this is (presumably) based on the exact same dataset as plot in Fig. 2a and b, i.e. explicitly state panel a is a linecut at zero bias of Fig 2a and panel b is calculated based on the data of Fig 2b.

Done

Results - continued

110: I kindly request the authors to show additional data. I find the amount of data presented in this part of the study is insufficient to support many of the conclusions made in this work. I would like to see the following data:

1. Bias dependent tunneling spectroscopy vs V_{gate} for all 3 TP's, at a range of B fields starting from 0 T, with at least a few magnetic fields before the supposed topological phase transition, and a sufficient density of data right before, during and after this possible phase transition. Such data should focus on the low energy behavior, i.e. the subgap states, so a bias range of $\sim \pm 0.5$ mV clearly suffices. The gate range should surely include the whole range covered in Fig 2, 3 and Extended data Fig. 5 and 6. If such data is not available, I am somewhat sceptical about the overall scientific merit of the work, and my final recommendation for acceptance or rejection in Nature Communications will largely depend on the completeness of these data. Please anticipate a request from my side for inclusion of such data in main text and/or extended data.

2. Bias dependent tunneling spectroscopy vs V_{gate} for all 3 TP's with a specific focus on ruling out / confirming localisation effects. I find this manuscript lacks a set of basic characterisation data of the device's behavior as a function of gate. I think this should be part of the Extended Data, and surely shared with me given my concerns. I am thinking of a large scale conductance map at $B = 0$ T, ranging in gate space from pinch off (can you actually pinch off your devices? If not, I expect repetitive or stable behavior below some negative gate voltage?), until saturation at high gate voltage, with a sufficient bias voltage range, i.e. order 5 mV. I would like to see such a dataset for all 3 TP's.

3. Bias dependent tunneling spectroscopy vs magnetic field - the main text device has 3 TP's, and I would like to see the equivalent data of Fig 3 (now shown only for TP1 and TP2) also for TP3. Already Fig 2c shows that non-trivial difference between TP 2 and TP3 are present at fixed gate voltage, and I think Fig 3 should simply be expanded from 6 till 9 panels, including TP3 (unless the data is somewhat 'boring', in which case this should still be shown as Extended Data).

- 1. We found that scanning the back-gate - back and forth - causes charge instabilities, and a reliable measurement is achieved only by stepping the back-gate in the negative direction (letting it rest) and scanning the magnetic field at each back-gate voltage step.*

A pity, but such is experimental life.

Please see below a sample of gate voltage dependence. The measurements were performed at $B=1.15T$. The left panel shows the conductance of the edge-TP1 as function of bias (x axis) and back-gate voltage (y axis). A robust ZBCP is present at a wide range of back-gate voltages.

The right panel shows the conductance of the bulk-TP2, measured with same parameters and at the same time as TP1. Here, we see a bulk gap is present at back-gate voltage range of $V_{BG} = -4.27V$ – $4.19V$. We note that here the bulk gap is biggest when the ZBCP at the edge is strongest (here at $V_{BG} = -4.24V$, slightly shifted than the maximal gap obtained in the ‘step-by-step’ measurement presented in the main text, at $V_{BG} = -4.22V$). The ZBCP gets weaker with increased back-gate voltage and closure of the bulk gap. The measurement shows a few charge instabilities, a phenomenon common in most of experimentalists in the field.

I don't understand why you don't put this in the Extended data and refer to it explicitly in the main text. Of course its noisy data due to charge switches but this is to be expected. It is interesting data somewhat supportive of a topologically non-trivial nature of the observations, although I personally don't think just this dataset resolves such a debate. To achieve that, you would, unfortunately, have to show a dataset of a larger extent, higher quality and higher internal consistency to convincingly argue that point. Nevertheless, I would show this data.

2. We added a measurement vs. gate voltage at zero magnetic field to the Extended Data (Fig. 15); however, we do not have a large-scale conductance map, but rather scattered measurements we performed around zero bias.
3. A complimentary figure of TP3 dependence on the back-gate voltage was added in the Extended Data Fig. 14.

110-113: Unnecessary in my opinion, I am in agreement, but you provide no argument why this holds and the sentence isn't very exact. I don't think anybody will really doubt your backgate is having a very significant effect - any conductance line trace at fixed V_{bias} while varying V_{gate} will show huge variations, proving this point. Mind you that in my opinion you cannot rule out that tunnel strength may fluctuate as a function of V_{gate} - see comment on line 345-346 of Methods section below, however, this is a different point.

Since the tunneling barrier of the TPs are made of Al oxide, with a barrier height of some 3eV, we do not see how the back-gate voltage will affect this barrier. We wish to keep this comment in with a change in the wording.

116: 'Being the largest observed gap...' and 119: '...and never reopens in the full range of magnetic field (Fig. 3c).': typical statements that needs backing up by additional data as requested under 1) above. You have the opportunity here to demonstrate the extend in V_{gate} of the ZBCP and reopened bulk gap, I strongly recommend doing so.

As stated above, the gap dependence on the back-gate voltage was measured and presented to Referee; however, not included into the paper due to instability issues.

See above.

122: Why do you have fluctuations in your effective g-factor? I am a bit sceptical about assigning effective g-factors to these states moving in B field. Clearly, they originate from a quasi-continuum at finite energy, where level repulsion causes a much 'slower' dispersion in B field. Often 'along the way' to zero energy other levels also disperse in B-field, which may lead to further distortions from a simple linear B field dependence. Lastly, how long a level 'sticks' to zero bias in magnetic field also heavily depends on repulsion from the nearby finite energy states. This influences assigning effective g-factors and points of 'gap closure' and 'gap reopening', and answering the naively binary question if a ZBCP is 'robust' or simply a cause of trivial ABS 'sticking' to zero bias for a while. Also, the coupling strength to the superconductor heavily influences all these behaviors. Ref 48 is quite illuminating in this respect and illustrates many of the points I am raising, you could consider citing this work in a more prominent spot in the text.

Fluctuations in the effective g-factor of sub-gap states occur when changing the electric field in the system (e.g. different back-gate voltage), as was shown in Vaitiekenas et. al., 2018. Indeed, there might be a more complex B-dependence which we didn't consider. Here we use a simple model, where we assume that the gap closes when the Zeeman energy is equal to the superconducting gap.

The estimate of the g factor in this way should gives the correct order of magnitude. We add a sentence addressing that issue in the text.

I follow your reasoning and I don't really object to what you do here, but we must all be well aware that the strength and usefulness of this simplistic argument is very limited in serving as a 'proof' for robustness of your ZBCP as a function of magnetic field, and the grand total of all these aspects (i.e. difficulty in proving robustness in gate voltage and magnetic field) leads me to the conclusion you cannot discriminate topologically non-trivial versus trivial causes for your observations; at best your results are suggestive of some non-trivial physics going on.

122-123: Another statement that needs extensive back-up by providing gate dependent

data as requested under 1) above.

As seen from the back-gate scans above, a ZBCP is present at the edge while either there is a reopened gap in the bulk, or a zero bias state present.

124-125: I appreciate the authors showing this behavior, as it constitutes a ‘negative’ result in the context of this paper’s main claim. Naturally, the question arises why any of the ZBCP’s behavior is strongly correlated with that of the ‘bulk gap’ measured at TP2. This connects to my overall comment of the possibility of probing individually localised states underneath each TP. A priori, this doesn’t necessarily mean all the reported behaviors of ZBCP and bulk gap are topologically trivial, but obviously a lot of nuancing seems very appropriated.

This issue had been addressed already above.

Figure 3 + caption

Please include data for TP3, unless it is identical to TP2 (Fig 2 doesn’t suggest so), in which case it should still be included as part of the Extended Data in my opinion. I suggest to follow the same ordering of panels as in Fig 2, i.e. each row corresponds to the same gate voltage. That makes it much easier to compare behavior at different TP’s for constant field. This would make Figure 3 a 9 panel figure. The authors could also consider combining Fig 2 and Fig 3 in a single Figure 2, and add a new Figure 3 with the complementary data of varying V_{gate} at constant B field. That would in my opinion be a much more comprehensive and consistent dataset.

A back-gate dependence of TP3 conductance vs. magnetic field was added to the Extended Data in Fig. 14.

As commented with Fig 2, consider placing something like ‘Edge’ , ‘Bulk - 1’ , ‘Bulk - 2’ above panels. I find several of the dotted ‘guide-to-the-eye’ lines obscuring nuances in the data, in particular in panel a around the crossing point of the states, for panel b around 1.3 T, and for panel f around 1.25T

A detail: In panel f, a switch is visible around 1.25T. Please share your opinion on the origin of this, and the occasional other switch visible throughout the data; is it an electron trap in the oxide (dis)charging, and if yes, it seems somewhat counterintuitive this happens as a function of magnetic field? How do you ensure such switches do not affect conclusions related to changing the chemical potential in the wire? Consider placing a short comment somewhere in the caption / main text/ methods section about the occasional occurrence of such switches (I congratulate the authors on achieving the level of stability displayed in the data throughout the paper, charge(?) switches seem rare which is great).

Lastly, I recommend to provide plots similar to those shown in Extended data Fig 3 c and d (taking my comments on those into account) and place these in the extended data.

The instabilities (can be of short and long time durations) are due to charging the

dielectric of the tunnel probe (Al_2O_3) and global gate dielectric (SiO_2), or at their interfaces. We are certain that during the measurements presented in Fig. 3 there were no switches (except from the one seen in Fig. 3f), so that regardless of the effect of such a charge fluctuations, the conclusions of this part hold.

Needless to say that such 'jumps' can may shift the chemical potential (locally or globally) in the device; preventing a reliable gate-scanning measurements.

Plots were added in the Extended Data Fig. 13.

Thank you for your clarification.

Results - continued

127-137: Although important from a viewpoint of completeness, I think this section and Figure 4 belong in the Extended Data. It is by now well established that the temperature dependence of the ZBCP does not aid in uncovering its origin, and follows a trivial pattern of suppression as a function of T. Similarly, the 'bulk gap' closure appear rather trivial to me. I encourage the authors to focus instead on the in my opinion much more important and interesting questions of gate dependence and comparing all the different types of behavior observed within the main device, and across all devices.

It is here from the point of completeness.

Figure 4 + caption

See comment on lines 127-137 above. Panel a: There is still a tiny ZBCP left at 195 mK. Could you show a few more traces at higher temperatures to prove the ZBCP really disappears? Panel b: I find it somewhat strange you don't show curves up to a similar T value as in panel a. Naturally, one now wonders if anything happens between the highest T value of 116 mK shown in panel b and the value of 195 mK shown in panel a. Lastly, it could be beneficial to duplicate the data as a colorplot, as different types of plotting draw attention to different types of features.

The highest temperature we measured with was 195mK. We believe that it is already clear that at this temperature the ZBCP nearly disappears. The features in bulk and edge were measured separately in this section (simultaneous measurement wasn't performed since one lock-in was used for accurate temperature measurement and the other for conductance of the TP). The desired temperature was achieved via heating of the mixing chamber by a constant current and waiting for full equilibration, which was difficult to reproduce (accurately) in different measurements at different times. Lastly, the bulk-TP was not measured at higher temperatures since the small 'shoulders' already disappeared at 116mK.

The data in the color-plot presented below was interpolated for clarity (linear interpolation of the data presented in Fig. 4 with steps of 2mK). It doesn't seem to add much information.

Thank you for your clarification.

Summary

144-145: language, ZBCP is already defined, not sure what journal policy recommends here. 'This is expected for the emergence...' seems better.

147-151: I feel this should be slightly expanded and rewritten based on the revisions of the paper I am suggesting.

Revised

Methods

Sample fabrication

This section is good as it is, except from one comment.

345-346: The effective tunnel barrier strength is not just dependent on the tunneling through the oxide (I indeed agree there's very little reason to expect such a tunnel strength will depend on the backgate voltage), but also depends on the exact probability density of the orbital nanowire wavefunction(s) near the tunnel barrier, which obviously will depend on the backgate voltage (although the effective 0V potential on the TP's may keep things fairly stable right underneath the TP's). In view of this I find this statement a bit misleading, it is anyway not necessary in this methods section.

Revised

Device parameters

354-355: If my earlier reasoning is correct, I would prefer defining E_z without the factor 2 in there.

See reply above

349-361: I find this estimate and the method used reasonable, and the lever arm found fits well with earlier experimental work on nanowire quantum dots (typically lever arm is of order 10% vs 1% in this work), especially considering the extra screening due the

aluminumshell.

363: The authors find a change in charge of the order of 1 electron per micrometer of nanowire per 10 mV change in backgate. This again emphasises my earlier point of the possibility of localisation effects being very important in this work: your TP's have a width of ~ 0.2 microns, i.e. pretty much of the order of 1 micron. This compares quite favorably to the number you estimate (factor 5 smaller), given the many assumptions that goes into the estimate.

365-373: I think this is better left out. By now it is well established that classical capacitance calculations (miserably) fail in these systems and the appropriate approach is to self-consistently solve the Schrodinger-Poisson equation. See Vuik et al., New Journal of Physics, Volume 18, March 2016 and consider citing this paper, as it is relevant to this work.

It does not hurt to leave it in and stress again the disagreement with reality.

Extended data

I have no further requests for data, and accept your preferred distribution of data over main text and extended data. In terms of remarks on the figures, I only missed a statement in each caption of a 'waterfall' plot of the stepsize in B field between each subsequent curve, please include that for completeness

Fig 1: see comments above.

The figure was revised.

Fig 2: Consider labelling panels with 'edge' and 'bulk'. Can you extend the plot by another ~ 200 mT? It seems the gap hasn't fully closed yet. Panel a: I find the 'guide-to-the-eye lines somewhat obscuring the details. Given that the tunnel barrier is so transparent, are you even sure there is a ZBCP? Maybe it is just two levels coming very close and appearing as a broad ZBCP? Lastly, I recommend also plotting the linetraces for this data.

Below attached the revised plots, extended up to $B=1.5T$. Line-traces were added to the plot for clarity. At the edge, a ZBCP appears ($B=0.5T$), then splits and recombines (at $B=1.15T$). We think that the data may be interpreted by stabilization of the topological state at $B>0.7T$, as for that value there is a ZBCP at the edge and a gap in the bulk.

However, the behavior at low field demands further interpretation, as it can result due to the high transparency of the edge probe (10 kOhm) and the short coherence length. However, following the Referee's remark we agree that the evidence for topological state

are not clear enough and therefore decide to remove the figure (see figure below) from the **Extended Data** in order to avoid confusion (as one can see we added a large number of figures to the Extended Data section).

402-404:

This is not a very strong statement, first you write that a ZBCP appears at 0.5 T, then that the bulk gap reopens at 0.6 T. I don't see a problem there? Having said that, I do indeed agree you may very well have had a different chemical potential for both tunnel probes.

Here we see the gap closing at $B=0.2T$. We expect that for a uniform chemical potential in both probes, a ZBCP should appear also at the edge at $B=0.2T$, and become stronger once the gap reopens. However, here the ZBP at the edge appears at higher field; hence, it is likely that the chemical potentials under the two TPs are different - but as noted above we decided to remove the figure, which may only confuse the reader.

More generally, is there more data on device 2, i.e. does it display other types of behavior for different gate voltages? It could be worth extending the figure with a few other types of behavior, if measured.

See reply above

I'm a bit agnostic towards including device 2 in the extended data or not. I think it's good in the sense that it shows reproducibility of your devices and set-up etc., I also think it doesn't add essential new information not captured by the main text device. I don't agree it's more confusing, I think your main text device is as 'confusing', and it's exactly this aspect of your results which prevents you from making solid conclusions towards observing topologically non-trivial physics.

Fig 3: Consider labelling panels a and b with 'edge' and 'bulk'. Panels c and d: I find the lines unclear because of all the individual dots. I am quite fine with just plotting a narrow line without individual datapoints. Also, I find the many colors not necessarily helpful, you could consider plotting it all in black and possibly indicate a few important traces by using a different color, such as onset of the ZBCP or point of gap closure / reopening.

Revised

417-421: Rather grand statement given the quality of the data. I fail to see: 1) why the ZBCP in panel a needs to be Majorana related, all I see is a gap closure pushing 2 pairs of ABS together to form a ZBCP right before superconductivity disappears. 2) a reopening of the gap in panel b, same arguments as for panel a). I find this a too large stretch. I would leave this dataset in the Extended Data, but give a more realistic down-to-earth description.

Revised

Same as for Fig 2 on device 2, is there more data that display other types of behavior for different gate voltages? It could be worth extending the figure with a few other types of behavior, if measured.

Yes, other types of behavior were measured. We have observed a ZBCP appearing at the edge with no bulk-gap closing and with a zero-bias state at the bulk at zero magnetic field. See added Fig. 6 and Fig. 9 in Extended Data.

Fig 4: Consider labelling panels a and b with 'edge' and 'bulk'.

Panel a): It seems a second pair of ABS 'merges' with the ZBCP at 0.9 T? Above that field value, the ZBCP is much broader, maybe there is a low energy 4 fold split conductance peak that gets washed out by the tunnel broadening?

panel b): Isn't there a weak ZBCP present at 1.2 T? Please comment.

The figure was revised. In both edge and bulk TPs there is an ABS that merges to a ZBCP (marked by dashed line in the positive bias side). Indeed, as the ABS merge at zero-bias, the ZBCP widens from $25\mu\text{eV}$ to $55\mu\text{eV}$, suggesting that the ABSs have an apparent finite energy due to level-repulsion smeared by tunnel-broadening and temperature.

Again, is there more data on device 1 after thermal cycling it, i.e. does it display other types of behavior for different gate voltages? Could you find a 'gap closure' and ZBCP appearance at similar fields as reported in Fig 2 and 3 main text? I recommend also plotting the linetraces for this data.

Yes, the thermally-recycled device also shows other types of behaviors, showing trivial ZBCPs at the edge with zero bulk-gap (added data in Extended Data Fig. 5 and Fig. 8, similar to the behavior in the Extended Data Fig. 4 and Fig. 7, respectively). TP3 was not measured in these cases.

Figs 2-4 form a useful additional dataset showing reproduction of the main features. In fact, I wouldn't mind extending the data in all 3 cases somewhat to cover a broader type of behaviors across the 3 devices.

Fig 5: I think this is an important dataset, that should be connected in some way to the results in the main text Figs 2 and 3. By changing the backgate voltage by a very small amount of 0.07V and 0.08V compared to the results in main text panels 3a,d and 3b,e respectively, suddenly this ZBCP at the edge became trivial. This really asks for some clarification. Is this the same feature as the one in the main text which is claimed to be non-trivial? If not, please proof this with experimental data, for example a continuous V_{gate} vs V_{bias} scan covering the whole range between main text Figs 2,3 and this figure, tracing the ZBCP. If it is the same feature, the authors should combine these data with those shown in the main text and make a single main text figure out of it. If it is not the same feature, I still strongly recommend to move this into the main text, as part of a separate figure that shows 'trivial' ZBCP's, with a corresponding discussion. Some more details: I think the data for TP3 should be included here as well (as with main text Fig 3). Label panels with 'edge', 'bulk 1', etc. Lastly typo in line 448, 'below' is what it should read.

The measurement in Fig. 5 (Fig. 4 in the revised version) shows a ZBCP appearing at the edge while the bulk gap closes and does not reopen (at least in our resolution) – at $V_{\text{BG}}=-4.29\text{V}$. This behavior is similar to the behavior presented at Figs. 3c & 3f in the main text with $V_{\text{BG}}=-4.13\text{V}$. It is trivial in the sense that there is no bulk-gap so the end-state is not topologically protected, exactly as in the case for $V_{\text{BG}}=-4.13\text{V}$ (Figs. 3c & 3f). Note, that on both sides of the 'sweet-spot', for higher and lower chemical potentials, the bulk-gap becomes smaller. Being similar to the data in Fig. 3 in the main text, it is better left in the Extended Data.

TP3 was not measured in this case.

Fig 6: Again, an important dataset. How does this connect (from an experimental viewpoint) to main text Figs 2 and 3, and the previous extended data Fig. 5? Do you have more data from neighboring gate values? If you can prove panel a) is a distinct feature from the one shown in main text Figs 2 and 3, it may still be worth showing these results in the main text as a separate figure. Isn't this another indication that what you measure at each TP isn't strongly related? Could you please include TP3's data as well?

The features that are shown here are different than what is shown in Figs. 2 & 3 – as the latter are in a different working point. Here, a ZBCP appears at the edge while the bulk shows a complicated magnetic field dependence of the ABS. We didn't measure continuously in the range $V_{\text{BG}}=-4.21\text{V}$ (Fig. 2) and $V_{\text{BG}}=-4.91\text{V}$ (Extended Data in Fig. 7). Here, we add, for the Referee, a measurement performed in the middle of this range, at $V_{\text{BG}}=-4.59\text{V}$ - showing a trivial behavior in the edge and in the bulk.

The edge shows a ZBCP throughout the full magnetic field range; hence, must be of trivial origin. TP3 was not measured in this data set.

Fig 7: I find this a very beautiful dataset, it should surely stay in the Extended data. Please consider my earlier remark, I don't really understand why you think this more opaque tunnel barrier is not suitable for edge tunneling spectroscopy. Please check the grammar in the caption there are several errors.

See an example of data taken with bulk-TPs with resistance in the Mega-ohm range. They barely show clear features below the SC gap. We could not sharpen or bring out better features.

Thank you for showing this.

List of requested changes, following line numbering of PDF file:

1-2: Title should be changed to reflect the inconclusive nature of the work. I suggest:

‘Concomitant opening of a bulk-gap with an emerging zero energy edge-state’.

which is both accurate and appropriate. Rest assured, the whole field will straightaway realize which conjecture you are implying here, yet you stay within the limits of what your experiment allows you to say. Whatever you come up with, I will have to refuse acceptance of any title in which you refer to your bulk-gap as ‘topological’ and your ZBCP as ‘Majorana’.

6-22: Abstract: It would be appropriate I feel if you actually do make the connection here that your data may represent an observation of topologically non-trivial physics (after all, it looks promising, and it can’t be ruled out), as I request you to not do so anywhere else in the manuscript apart from the discussion. It would also put the statement you’re currently making on the importance of this type of measurement in its appropriate context. I therefore suggest for line 16-22:

...‘For a particular gate-tuning of the chemical potential in the wire and a Zeeman field (parallel to the wire), we observed a closing of the superconducting bulk-gap followed by its reopening simultaneously with the appearance of a ZBCP at the edge’, **suggestive of the occurrence of a topologically non-trivial phase with associated Majorana zero-mode.** ‘Yet, we found that a ZBCP could also be observed with different tuning parameters without an observed reopening of the bulk-gap. This demonstrates the importance of simultaneous probing of the bulk and edge when searching for a Majorana edge-state.’

78: ‘...to most previous configurations,...’ I think you should refer to those experiments explicitly here, I think you referred to them anyway earlier, but the reader needs to know which experiments you’re talking about here.

113/114: the word topological needs to be omitted here, and instead, a more neutral and interpretation free phrasing should be chosen. I suggest something along the lines of: **‘Overall, the closure and reopening of the bulk gap happens for $B > 1.4$ T for all TP’s.’**

114/115: I really don’t get the need for this statement at this stage, nor anywhere in the manuscript as a matter of fact, as you anyway can’t be sure you’re measuring a topologically non-trivial phase. Even if you could be sure you were, this statement is rather random and stand alone, without any theoretical or experimental back-up currently – I don’t ask you to do so - I think you should just delete that sentence.

117: ‘topological’ should be omitted without replacement, or be replaced by **‘bulk’**

119/120: You could make this statement a bit clearer and tighter, e.g.: ‘The global gate voltage is not expected to affect the tunneling probability of the **metal** probes since their coupling is determined by a high-barrier Al-oxide. Hence, the observed dependence on the gate voltage

reflects its effect on the chemical potential in the wire.’ As you already say ‘is not expected’ in the first sentence, the words ‘it is fair to assume’ are unnecessary.

123: ‘trivial’ should be omitted

124/125: Make clear you speculate about a possible explanation in terms of non-trivial physics. Can easily be achieved by my following suggestion:

‘Being the largest observed gap, it **may be** that the Fermi energy is placed very close to the ‘sweet spot’ **for the formation of a possible topologically non-trivial phase.**’

128-130: No objection against the usage of the word ‘topological’ here, but consider replacement with ‘**topologically non-trivial**’

136/137: Here missing commas and unnecessary parentheses are actually hindering easy understanding of the statement. Hence I suggest: ‘Alternatively, as the bulk gap does not reopen in Fig. 3f, the ZBCP in Fig. 3c is likely a ‘trivial’ ZBCP.’

141: emphasize speculative character: ‘**...of a possible topologically non-trivial state.**’ or ‘**...of a possible non-trivial state.**’

158/159: emphasize speculative character: ‘...by its reopening (as a **possible** ‘topological gap’),...’

161: Emphasize the sensitivity to gate voltage here: ‘...The **sensitivity** of the ‘bulk-gap’...’

163: Emphasize the tiny change in gate voltage required to achieve this: ‘...we find that for other **nearby** values of the chemical potential...’

164-167: Bit inelegant, and as stated earlier, I request a more direct statement here. A suggestion for an alternative:

‘This sensitivity to gate voltage suggests localization effects are important in the experiment; these are either caused by intrinsic disorder in the device, or by tunnel probe induced potential fluctuations, and further theoretical modelling is required to shed light on this. Although suggestive, this sensitivity hinders the conclusion of a topologically non-trivial phase being the cause of the observed concomitance of bulk gap closure and reopening and ZBCP. The observed occasions of trivial ZBCP’s closely resembling Majorana ZBCP’s, as revealed by simultaneous probing of the bulk and edge, clearly underline the importance of this technique, and bear relevance to all observed ZBCP’s up to date.’

183/184: See the comment in my initial response – just share it as a supplementary file and no need for this. The phrasing ‘reasonable request’ is somewhat odd? Shouldn’t the data always be available upon request?

334: '(as a possible topological gap)...

349: replace 'topological' by 'bulk', i.e. '...with bulk gap $E_g \sim 25 \mu\text{eV}$.'

407: In the figure, place the words trivial and topological between quotes to indicate this is speculative.

412: '...At the possible topologically non-trivial region, the...'

413: Delete 'topological', i.e. '...where the gap is maximal...'

443: Delete 'topological', i.e. '...with a maximal gap size of...'

Editor

We have accepted the vast majority of changes suggested by the Referees and hope that now the paper is suitable for publication in Nature Communications. Below attached our answers to the Referees in *italic blue*.

Reviewer #1

"More importantly, Fig. 3c clearly shows a robust ZBA after the gap closing, something unexpected in the bulk. Can the authors comment on this? "

The answer to this point is that owing to the finite energy resolution, of the order of $\sim 25\mu\text{eV}$, a very small gap may appear as a conductance peak. My question is then how do the authors distinguish trivial parity crossings from robust ZBAs? More specifically, what is the criterion that allows to interpret a peak like the one shown in Fig. 2a in terms of a robust ZBA and not some finite-energy crossing blurred by energy resolution? I think this important point deserves further discussion (both in the answer to my comment and in the manuscript).

Our observations, namely, the closure and reopening of the gap in the bulk and the robust zero bias peak at the end are consistent with the emergent Majorana zero mode in a topological superconductor. We cannot rule out scenarios with accidental finite energy states at the end of the wire with splitting that is below our energy resolution (and is not sensitive to the gate potential). We emphasize this point in the manuscript (second paragraph in the results).

Referee 3

We accepted many of your suggestions and hope that now the paper presents the data in a clear and "honest" way. We thank you again for investing the time to read our paper thoroughly and improve it significantly.